# Efficient precise in vivo base editing in adult dystrophic mice

Li Xu[1,3], Chen Zhang[1,3], Haiwen Li[1], Peipei Wang [1], Yandi Gao[1], Nahush A. Mokadam[1], Jianjie Ma [1], W. David Arnold[2] & Renzhi Han [1✉]

Recent advances in base editing have created an exciting opportunity to precisely correct disease-causing mutations. However, the large size of base editors and their inherited off-target activities pose challenges for in vivo base editing. Moreover, the requirement of a protospacer adjacent motif (PAM) nearby the mutation site further limits the targeting feasibility. Here we modify the NG-targeting adenine base editor (iABE-NGA) to overcome these challenges and demonstrate the high efficiency to precisely edit a Duchenne muscular dystrophy (DMD) mutation in adult mice. Systemic delivery of AAV9-iABE-NGA results in dystrophin restoration and functional improvement. At 10 months after AAV9-iABE-NGA treatment, a near complete rescue of dystrophin is measured in $mdx^{4cv}$ mouse hearts with up to 15% rescue in skeletal muscle fibers. The off-target activities remains low and no obvious toxicity is detected. This study highlights the promise of permanent base editing using iABE-NGA for the treatment of monogenic diseases.

[1] Division of Cardiac Surgery, Department of Surgery, The Ohio State University Wexner Medical Center, Columbus, OH, USA. [2] Department of Neurology, The Ohio State University Wexner Medical Center, Columbus, OH, USA. [3] These authors contributed equally: Li Xu, Chen Zhang. ✉email: renzhi.han@osumc.edu

Duchenne muscular dystrophy (DMD) is a fatal genetic muscle disease affecting approximately 1 in ~5000 male births worldwide[1], which is caused by mutations in the *DMD* gene[2]. Most of the *DMD* mutations are due to deletions or duplications with over 500 point mutations accounting for ~10% of the cases[11,12]. DMD codes for the dystrophin protein[3], a cytoskeletal protein that functions in the muscle force transmission and sarcolemmal stability of muscle fibers[4]. Loss of dystrophin leads to progressive muscle weakness and wasting, loss of ambulation, respiratory impairment, cardiomyopathy, and eventual death[5].

Previous studies showed that exon deletion through clustered regularly interspaced short palindromic repeats (CRISPR) genome editing can restore dystrophin expression and function in dystrophic animals and cells[6–20]. Although promising, this strategy raises potential safety concerns as it relies on the repair of the double-strand DNA break (DSB) created by CRISPR/Cas9[21–23], which may cause unwanted large deletion and even DNA rearrangement[24–26]. Through fusing the Cas9 nickase with nucleobase deaminases (e.g., cytidine or adenine deaminase), a new paradigm-shifting class of genome editing technology, termed "base editors", have recently been developed[27–29]. DNA base editors, via catalyzing the conversion of one base to another, directly and precisely install point mutations into chromosomal DNA without making DSBs. Therefore, base editing can be developed as a promising therapeutic strategy to correct genetic diseases without DNA cleavage. In particular, the adenine base editors (ABEs) show remarkable fidelity in mouse embryos and plants as compared to cytosine base editors (CBEs)[30,31], making them highly attractive in therapeutic development. Moreover, nearly half of the point mutations causing human diseases are G-to-A or C-to-T[29], highlighting the potential of ABEs in correcting a large number of human diseases. In particular, 174 out of 508 pathogenic point mutations for DMD are due to G:C to A:T conversion (Supplementary Table S1), which could potentially be targeted by ABE editing.

A recent study showed that in vivo base editing can correct a custom-made mouse model of DMD[32], which carries a nonsense mutation in exon 20 with a classical 5'-TGG protospacer adjacent motif (PAM) sequence in the noncoding strand for recognition by the Cas9 from *Streptococcus pyogenes* (SpCas9). In silico analysis of the ClinVar database showed that about 42.8% of the 53469 human disease-causing mutations could be potential targets for base editing correction; however, the majority (~72.4%) of these potential targets could not be suitable for SpCas9 base editing due to the lack of the 5'-NGG PAM sequence within the suitable distance from the mutations. Several variants of SpCas9 have recently been engineered with relaxed PAM (such as xCas9-3.7[33], SpCas9-NG[34], and ScCas9[35]) and non-G PAM[36,37]. These enzymes greatly increase the target scope for correcting human mutations. However, their performance to correct genetic mutations in preclinical animal models remains to be determined. Here we explore the feasibility and long-term efficacy of correcting a commonly used mouse model of DMD, *mdx^{4cv}* mice[38], using NG-targeting base editors.

## Results

**In vitro reporter assay demonstrates the feasibility to correct the *mdx^{4cv}* mutation using ABE-NG.** The *mdx^{4cv}* mouse carries a premature stop codon (CAA-to-TAA) in the exon 53 of the *Dmd* gene[38], which disrupts the expression of dystrophin and leads to the development of muscular dystrophy. Targeting the noncoding strand with ABEs could potentially correct this nonsense mutation. However, in the noncoding strand, there is a lack of 5'-NGG sequence at the downstream of this mutation within

the suitable editing window. But a 5'-TGT PAM is present with the mutated A located at position 4 in the guide RNA (gRNA) (Fig. 1a), making it feasible to correct the stop codon with the NG-targeting base editors in this widely used mouse model of DMD. We first constructed a reporter plasmid[39] with the targeting sequence from the *mdx^{4cv}* mice (Fig. 1b). The nonsense mutation in the *mdx^{4cv}* targeting sequence disrupts the expression of downstream EGFP and successful editing of the nonsense mutation is indicated by the restoration of EGFP expression. As shown in Fig. 1c, transfection with the reporter alone resulted in minimal background fluorescence. Similarly, co-transfection with the reporter, *mdx^{4cv}*-gRNA and ABEmax failed to restore EGFP expression. However, ABE-NG (based on SpCas9-NG) successfully restored EGFP expression in this reporter assay. In contrast, ABE-x (based on xCas9-3.7) was found to be less efficient in restoring EGFP expression even though xCas9-3.7 was also engineered to target 5'-NG PAM, consistent with previous reports that xCas9-3.7 is generally less efficient than SpCas9-NG[34,40]. FACS analysis showed that ABE-x and ABE-NG restored EGFP expression in 10% and 20% cells, respectively (Fig. 1d, e). These in vitro studies showed that ABE-NG could potentially correct the nonsense *mdx^{4cv}* mutation.

**Modified variants of ABE-NG improve the editing efficiency and specificity.** The relatively low efficiency of ABE-NG, together with the recently reported off-target RNA editing activity[41,42], prompted us to re-design ABE-NG in order to improve the editing efficiency and specificity. First, we reasoned that the targeting efficiency of ABE-NG at the sites with 5'-NG PAM could be improved by optimizing the PAM-interacting domain. We hypothesized that the targeting property of ABE-NG can be modified by combining the mutations in SpCas9-NG (R1335V/L1111R/D1135V/G1218R/E1219F/A1322R/T1337R) with other mutations designed to target different PAM sequences such as those in xCas9(3.7) (A262T/R324L/S409I/E480K/E543D/M694I/E1219V), VQR (D1135V/R1335Q/T1337R), VRER (D1135V/G1218R/R1335E/T1337R) and the loop sequence in ScCas9 (amino acids 367-376). We generated seven new ABE variants with different combinations of the aforementioned mutations (details are provided in Supplementary Table S2) and compared their base editing activities at five different loci with those of ABE-NG and ABEmaxSC. While all variants except ABE-NGC (containing all NG mutations plus R1335E) performed similarly at the NGG site (Fig. 2a), we observed that ABE-NGA (carrying all NG mutations plus R1335Q) improved the editing efficiency at the NG sites as compared to ABE-NG (Fig. 2b–e), with an average efficiency of 41.9 ± 9.1% for ABE-NG and 51.3 ± 9.9% for ABE-NGA ($p = 0.007$), (Fig. 2f). Interestingly, ABE-NGA and ABE-NGX-NGC (carrying the xCas9(3.7) mutations, NG mutations and R1335E) worked equally well at the NGC site (Fig. 2c). The ABE-NGX variant carrying both the xCas9(3.7) mutations and ABE-NG mutations was previously reported to have the broadest targeting scope in plants[43]. However, the efficiency of ABE-NGX at the NG sites was not significantly different from that of ABE-NG ($p = 0.35$) but significantly lower than that of ABE-NGA ($p = 0.04$) (Fig. 2f). Since ABE-NGA is generally superior to other variants tested at NG sites, we chose ABE-NGA for further in vitro and in vivo studies.

Previous studies showed that the deaminase domain in the ABEs could elicit transcriptome-wide RNA off-target editing activity[41,42], and that the off-target RNA editing activity can be eliminated by removing the WT ecTadA domain[44] and mutating the evolved ecTadA domain[41,42,44]. We replaced the dimeric adenine deaminase domain (ecTadA-ecTadA*) in ABE-NG with the originally evolved ecTadA* monomer or its high-fidelity

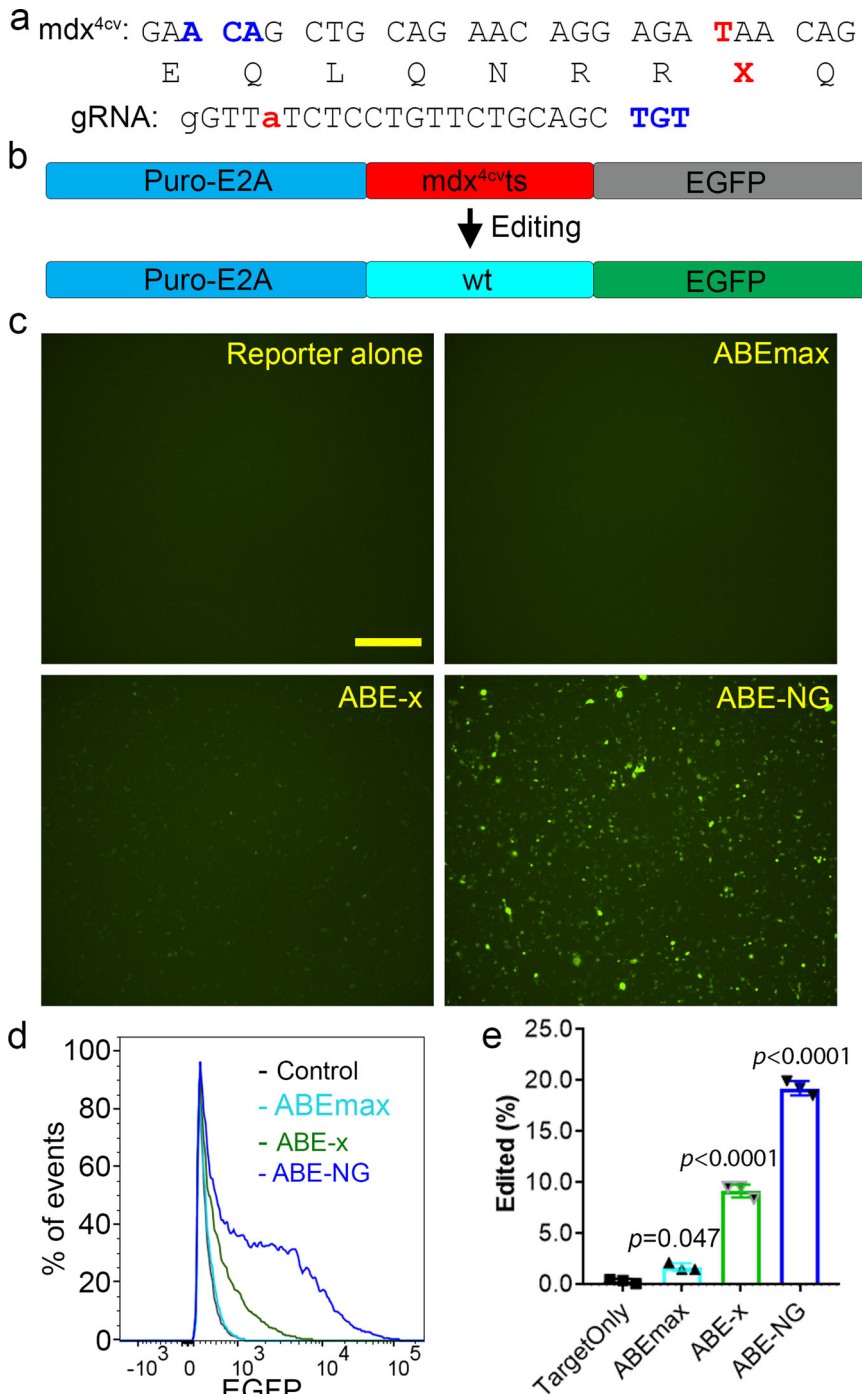

**Fig. 1 In vitro studies of mdx$^{4cv}$ mutation correction using ABE-NG. a** Genomic DNA, encoded amino acids and guide RNA with PAM (highlighted in blue) sequences at the stop codon mutation site (red). **b** The reporter construct contains a puromycin resistance cassette fused with E2A peptide, *mdx$^{4cv}$* target sequence and ATG-removed EGFP. Correction of the stop codon within the target sequence would allow EGFP expression. **c** Fluorescence microscopy images of HEK293 cells transfected with reporter alone, or reporter, gRNA and one of the base editors (ABEmax, ABE-x, and ABE-NG). Scale bar: 500 μm. **d**–**e** Flow cytometry analysis of EGFP expression in HEK293 cells transfected as described in (**c**). $n = 3$ wells/group; one-way ANOVA with Turkey's multiple comparisons test. Data are mean ± SD.

version (ecTadA*-V82G) (Fig. 3a) in order to minimize the off-target RNA editing activity. We confirmed that the miniABE-NG (the monomeric TadA* fused with SpCas9-NG nickase) had an increased efficiency at the *mdx$^{4cv}$* target site as compared to ABE7.10-NG ($p < 0.0001$, Fig. 3b). However, the on-target DNA editing activity of miniABE(V82G)-NG was remarkably reduced by over 50% when compared to ABE7.10-NG ($p < 0.0001$, Fig. 3b). We then attempted to improve the on-target DNA editing

efficiency of the high-fidelity miniABE(V82G)-NG without compromising its low off-target RNA editing activity. The V82G is one of the 26 amino acid residues in ecTadA that likely reside near the enzymatic pocket around the substrate tRNA, inferred from the *S. aureus* TadA-tRNA co-crystal structure[44]. We reasoned that the V82G mutation does not only affect the non-specific affinity to RNA substrates, but may also reduce its affinity to the DNA substrates. We noticed that the A56G

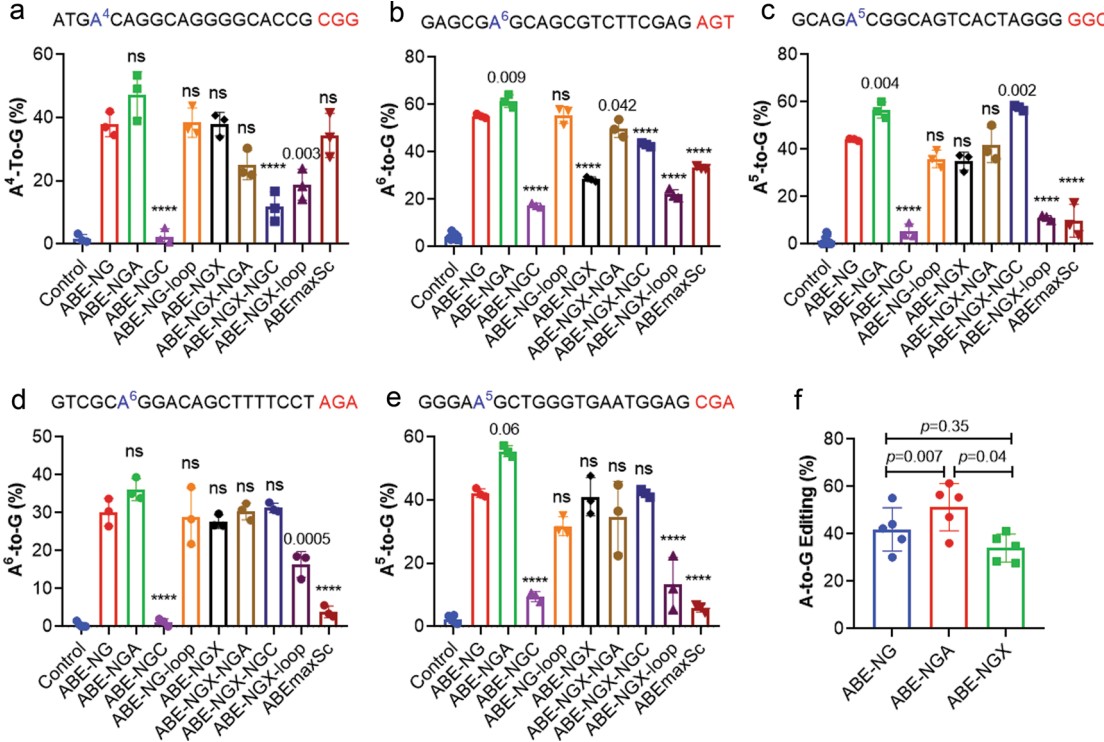

**Fig. 2 Optimization of the PAM-interacting domain to improve the editing efficiency of ABE-NG at the NG sites. a–e** Measurements of the base editing efficiencies of different ABE variants on five different sites with NG PAM. The gRNA sequences are listed on the top of each graph with the PAM sequences in red and the target adenines in blue. *ns* not significant; ****$p < 0.0001$; one-way ANOVA with Turkey's multiple comparisons test to compare with the ABE-NG group. $n = 3$ wells/group. Data are mean ± SD. **f** Comparison of the editing efficiencies of different ABE variants at all five NG sites. $n = 5$ different sites/group; average of three different wells per site; repeated measures one-way ANOVA with Turkey's multiple comparisons test. Data are mean ± SD.

mutation (which is also inferred to lie near the enzymatic pocket around the tRNA substrate) had higher on-target DNA editing activity without affecting the off-target RNA editing activity as compared to miniABEmax in the original publication by Grünewald et al.[44]. We hypothesized that installing the A56G mutation into miniABE(V82G)-NG may improve its on-target DNA editing activity without compromising its off-target RNA editing profile. Indeed, we observed that adding the A56G mutation into miniABE(V82G)-NG (named miniABE(GG)-NG) completely restored its on-target DNA editing activity (Fig. 3b).

We then used RNA-seq to compare the transcriptome-wide off-target RNA editing activities of miniABE(GG)-NG to other ABE variants in mouse Neuro-2a cells. We performed these studies in triplicates. Edited RNA adenines were identified from RNA-seq experiments as previously described[42] by filtering out background editing observed with read-count-matched controls. Consistent with the previous report[44], miniABE-NG induced much higher numbers of adenine editing as compared to miniABE(V82G)-NG ($p = 0.003$, Fig. 3c), and the A56G mutant was similar to miniABE-NG ($p = 0.97$). When compared to miniABE-NG, the GG variant maintained significantly reduced off-target RNA editing activities ($p = 0.025$), which was not significantly different from miniABE(V82G)-NG ($p = 0.45$). To further verify the low off-target RNA editing activity of miniABE (GG)-NG, we compared the A-to-I RNA editing at four previously characterized loci in human cells, which were shown to be highly modified by ABEmax[44,45]. We amplified and sequenced the RT-PCR amplicons. As expected, transfection of HEK293 cells with miniABE-NG induced high levels (30–57%) of A-to-I RNA editing in all these transcripts (Fig. 3d); however, such high A-to-I RNA editing was essentially eliminated in cells

transfected with miniABE(GG)-NG (Fig. 3d). Taken together, our results showed that miniABE(GG)-NG does not only have increased on-target DNA editing activity, but also inherits the low off-target RNA editing activity of miniABE(V82G)-NG. Hereafter, the improved ABE-NG carrying the miniABE(GG) domain and Cas9-NGA nickase was referred to as iABE-NGA.

Two groups recently reported a new generation of ABEs through directed evolution, namely, ABE8s (such as ABE8.17 and ABE8.20)[46] and ABE8e[47]. To directly compare miniABE(GG) with ABE8.17, ABE8.20, and ABE8e, we fused each of them with SpCas9-NG and tested their activities for editing the *mdx[4cv]* target site using the reporter assay in Neuro-2a cells. All these editors showed above 60% editing efficiency with the ABE8e-NG exhibiting the highest activity (Supplementary Fig. S1a). We predicted that ABE8e may have also increased bystander activity than miniABE(GG). Since the *mdx[4cv]* target site has no extra adenine within the editing window, we compared the bystander editing activity of ABE8e and miniABE(GG) by testing their performance to edit a nonsense mutation in human *DYSF* gene (encoding dysferlin) that causes limb girdle muscular dystrophy type 2B[39]. The target $A^6$ was edited with ~58% and 88% efficiency by iABE-NGA and ABE8e-NG ($p < 0.0001$), respectively (Supplementary Fig. S1b). The two bystander adenines at positions 8 and 11 were also edited at substantially higher rates by ABE8e-NG than miniABE(GG)-NG ($A^8$: 59.2 ± 1.0% vs 8.9 ± 0.7%, $p < 0.0001$; $A^{11}$: 13.6 ± 0.4% vs 0.4 ± 0.4%, $p < 0.0001$). The ABE8e with V106W mutation still displayed very high bystander editing activity at positions 8 and 11 ($A^8$: 48.0 ± 1.5%, $p < 0.0001$; $A^{11}$: 7.2 ± 1.2%, $p < 0.0001$). Thus, we chose miniABE(GG) for the in vivo studies due to its high efficiency and relatively high precision.

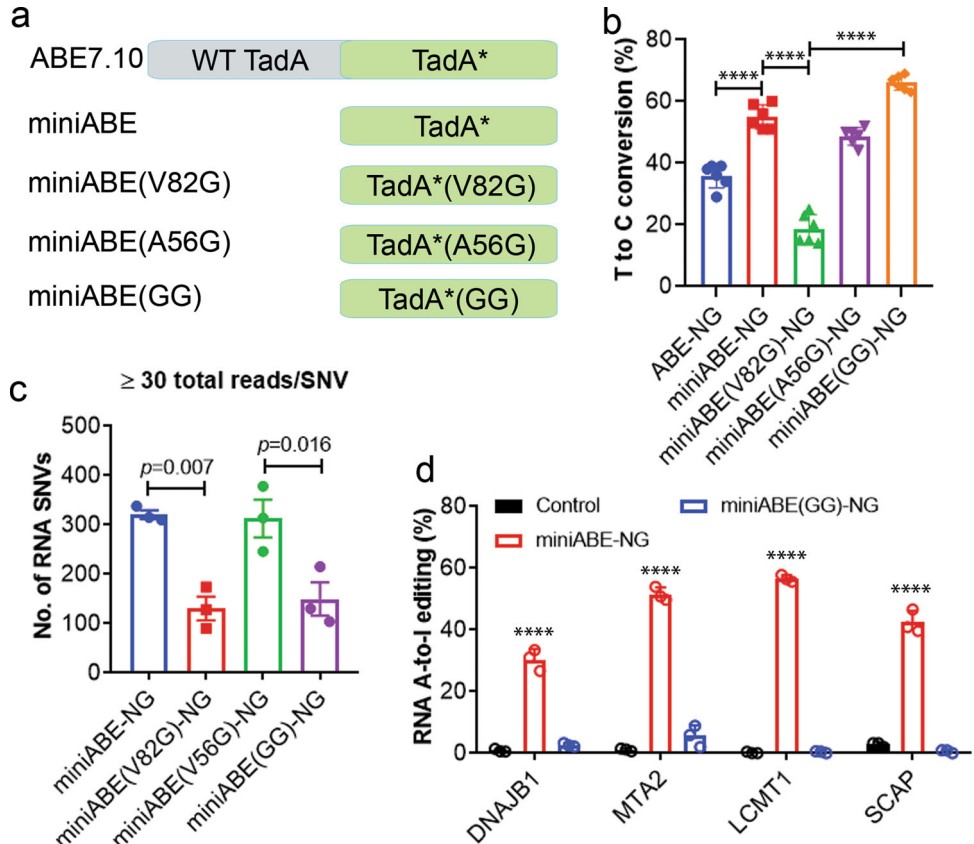

**Fig. 3 Re-engineering the adenine deaminase domain to improve the efficiency and specificity of ABE-NG. a** Schematics of the adenine deaminase domain used in ABE-NG editors. **b** Quantification of the editing efficiency of different ABE-NG variants with modified TadA* domain at the $mdx^{4cv}$ target site. $n = 6$ wells/group; ****$p < 0.0001$; one-way ANOVA with Turkey's multiple comparisons test. Data are mean ± SD. **c** The number of off-target RNA editing events in Neuro-2a cells transfected with different ABE-NG variants. $n = 3$ wells/group; one-way ANOVA with Turkey's multiple comparisons test. Data are mean ± s.e.m. **d** Quantification of the off-target RNA editing (A-to-I) activities on four RNA adenines previously identified as being efficiently modified by ABEmax in HEK293 cells. $n = 3$ wells/group; ****$p < 0.0001$; one-way ANOVA with Turkey's multiple comparisons test. Data are mean ± SD.

**Intein-split allows efficient assembly of full-length ABE-NG and editing**. The large size of the ABE-NG and other base editors poses a major challenge for viral packaging and in vivo delivery. A dual trans-splicing adeno-associated virus (AAV) approach was previously used to deliver ABE[32] and a dual protein trans-splicing (PTS) approach using the split-intein moiety from *Nostoc punctiforme* (*Npu*)[48] was used to deliver CBE[49]. We attempted to adopt the PTS approach to deliver ABE. In our initial preliminary experiments, the ABE was split between the ecTad-ecTadA* and the Cas9 nickase with *Npu* intein moieties, and this split rendered low editing efficiency (Supplementary Fig. S2). To improve the editing efficiency of the split ABE, we chose the amino acid position 573 and 574 of the Cas9 nickase as the splitting site because previous studies showed that 573/574 split Cas9 exhibited near the full-length Cas9 activity[48]. Moreover, split at this site would produce a roughly equal size of the two halves for AAV packaging (Fig. 4a). We reasoned that the split ABE could be further improved by using inteins with fast rate of PTS. We selected two inteins with the remarkably fast rate of PTS: Cfa ($t_{1/2}$ = 20 s at 30 °C)[50] and Gp41-1 ($t_{1/2}$ = 5 s at 37 °C)[51], which are ~2.5-fold and ~10-fold faster than the rate reported for the *Npu* DnaE intein ($t_{1/2}$ = 50 s at 37 °C)[48], respectively. Transfection of both split versions into HEK293 cells resulted in robust expression of full-length ABEs as detected by the anti-Cas9 antibody (Fig. 4b), although the expression level was generally lower than the ABEmax but higher than the original ABE7.10. Co-transfection with the split ABE-NG, $mdx^{4cv}$-gRNA and the $mdx^{4cv}$ reporter restored EGFP expression to a similar level as the

full-length ABE-NG (Fig. 4c). There was no significant difference between the Cfa and Gp41-1 intein splits ($p = 1.0$, Fig. 4d). We chose the Gp41-1 version for further studies.

Recently, the Liu group showed that the Npu intein split of ABE worked well in vitro and in vivo[52]. We directly compared the Gp41-1 split and Npu split. While both the Gp41-1 split and Npu split allowed the assembly of full-length iABE-NGA, Western blotting analysis showed that the Gp41-1 split rendered ~4-fold more full-length iABE-NGA protein than the Npu split ($p < 0.0001$, Fig. 4e, f). The assembly efficiency (as measured by the percentage of the full-length band) of the Gp41-1 split was about 69%, while the Npu split resulted in only 21% ($p < 0.0001$, Fig. 4g). To further compare the editing efficiency of the Gp41-1 split and Npu split, we quantified the T-to-C conversion of the $mdx^{4cv}$ stop codon in Neuro-2a cells using the reporter assay. As compared to the full-length iABE-NGA, the Gp41-1 split and Npu split retained about 85.5% and 78.8% of its activity, respectively (Fig. 4h). The difference in the editing efficiency between the Gp41-1 split and Npu split was small (56.4 ± 2.0% vs 52.0 ± 1.5%) but statistically significant ($p = 0.0009$) (Fig. 4h). In our Gp41-1 split and Npu split, each half carries a U6-gRNA expression cassette, while only the C-terminal half of the Npu split reported by the Liu group carries the U6-gRNA expression cassette[52]. To test if the double U6-gRNA cassettes have higher editing activity than a single U6-gRNA cassette, we removed the U6-gRNA cassette from the N-terminal construct of our Npu split, which resulted in lower editing ($p < 0.0001$, Fig. 4h), consistent with previous reports that the gRNA dosage is

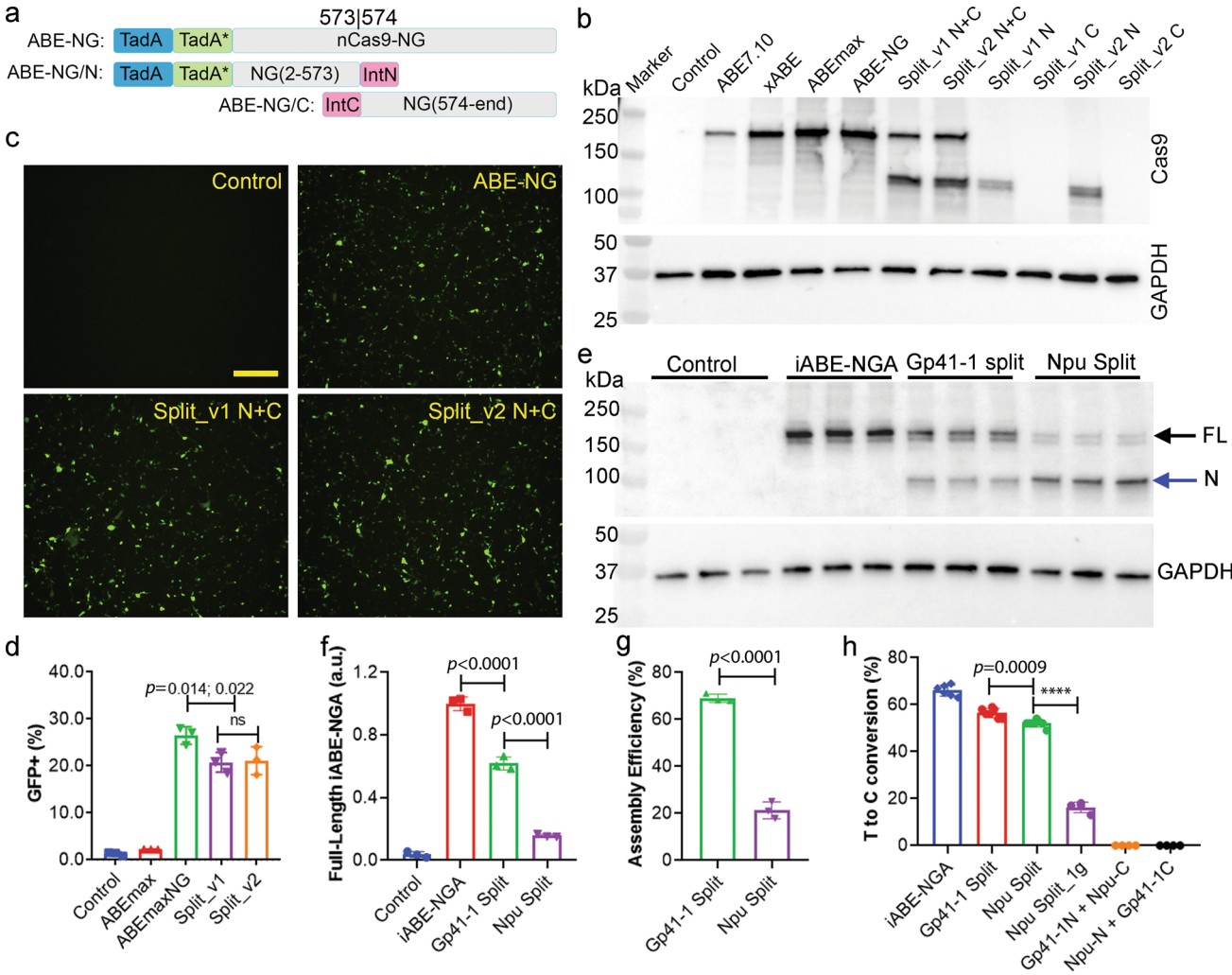

**Fig. 4 Intein-mediated assembly of full-length ABE-NG. a** Schematics of the intein split ABE-NG. The N-terminal and C-terminal intein sequences reconstitute the full-length protein when co-expressed within cells. **b** Western blot analysis of HEK293 cell lysates transfected with different versions of ABEs. **c** Fluorescence microscopy images of HEK293 cells transfected with reporter alone, or reporter, gRNA and one of the base editors (ABE-NG, split_v1 N + C or Split_v2 N + C). Scale bar: 500 μm. **d** Flow cytometry analysis of EGFP expression in HEK293 cells transfected as described in (**c**). $n = 3$ wells/ group; *ns* not significant; one-way ANOVA with Turkey's multiple comparisons test. Data are mean ± SD. **e** Western blot analysis of HEK293 cell lysates transfected with full-length iABE-NGA, Gp41-1, or Npu split of iABE-NGA. FL, the full-length iABE-NGA band; N, the N-terminal fragment of the iABE-NGA. **f** Densitometry quantification of the Western blot data shown in (**e**). $n = 3$ wells/group; one-way ANOVA with Turkey's multiple comparisons test. Data are mean ± SD. **g** The assembly efficiency of the Gp41-1 and Npu split of iABE-NGA (defined as the percentage of the full-length iABE-NGA bands). $n = 3$ wells/group; one-way ANOVA with Turkey's multiple comparisons test. Data are mean ± SD. **h** Quantification of the editing efficiency of full-length iABE-NGA, Gp41-1 split, and Npu split of iABE-NGA at the $mdx^{4cv}$ target site. Npu Split_1g is same as Npu Split except that only the C-terminal construct carries the gRNA. $n = 6$ wells/group for iABE-NGA; $n = 7$ for Gp41-1 Split and Npu Split; $n = 3$ wells/group for Npu Split_1g, Gp41-1N + Npu-C, and Npu-N + Gp41-1C; ****$p < 0.0001$; one-way ANOVA with Turkey's multiple comparisons test. Data are mean ± SD.

important for efficient Cas9-mediated editing[16,53]. Moreover, to test the specificity of intein-mediated assembly of iABE-NGA, we swapped the N and C-terminal fragments of the Gp41-1 and Npu splits, and observed no editing (Fig. 4h), suggesting that the intein-mediated protein splicing and assembly of full-length iABE-NGA are required for efficient editing.

**Systematic delivery of AAV9-iNG leads to widespread dystrophin restoration.** We packaged the two Gp41-1 intein split halves of the iABE-NGA into AAV9 (hereafter referred to as AAV9-iNG) and tested if in vivo delivery of iABE-NGA could correct the mutation in $mdx^{4cv}$ mice. A truncated MHCK7 promoter was used to drive the expression of two halves of iABE-NGA. A preliminary testing of two dosages (a total of $5 \times 10^{13}$ or $1 \times 10^{14}$ vg/ kg, 1:1 of the N and C-terminal half) showed that the higher dose

appeared to increase the dystrophin-positive myocytes in the $mdx^{4cv}$ mouse heart (Supplementary Fig. S3). We thus chose the higher dose ($1 \times 10^{14}$ vg/kg, 1:1 of the N and C-terminal half) for the rest of the study. In addition, our preliminary study also showed that injection of AAV9-iNG carrying a non-targeting gRNA failed to induce dystrophin rescue (Supplementary Fig. S3).

We treated a cohort of nine $mdx^{4cv}$ mice with AAV9-iNG (a total of $1 \times 10^{14}$ vg/kg, 1:1 of the N and C-terminal half) through a single tail vein injection at 5 weeks of age. A subset of the mice was sacrificed at 5 weeks after AAV9-iNG administration. Dystrophin was found to be widely rescued in $mdx^{4cv}$ heart (Fig. 5a and Supplementary Figs. S4–10). Quantification of the entire heart sections showed that $41.9 \pm 10.5\%$ cardiomyocytes of $mdx^{4cv}$ mice became dystrophin positive at 10 weeks of age after systematic AAV9-iNG treatment ($N = 5$) while the control

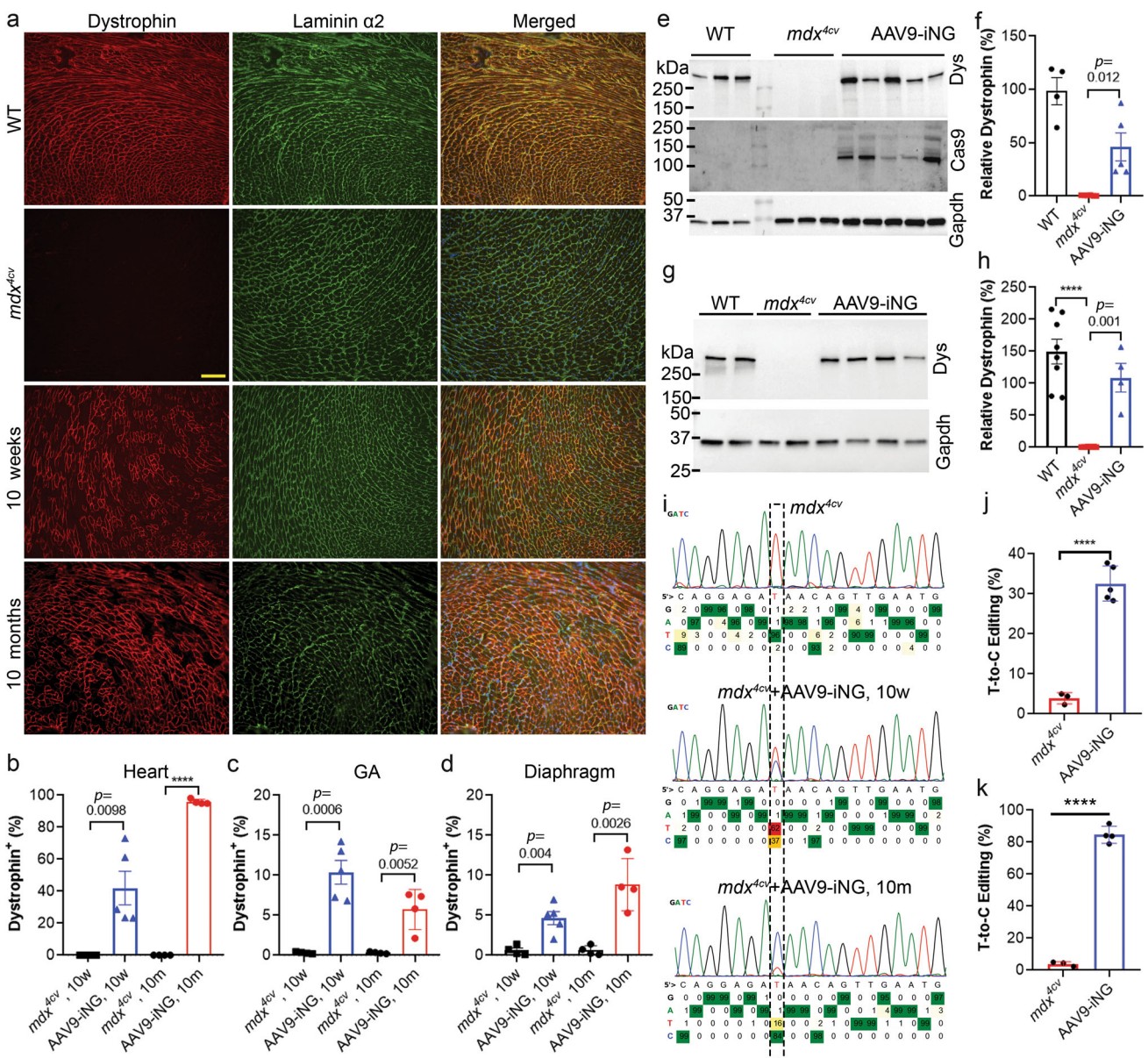

**Fig. 5 Widespread restoration of dystrophin expression in *mdx$^{4cv}$* mice following systemic administration of AAV9-iNG. a** Dystrophin and laminin-α2 co-immunostaining of heart sections from WT and *mdx$^{4cv}$* mice (10 weeks or 10 months of age) with or without tail vein injection of AAV9-iNG (a total of $1 \times 10^{14}$ vg/kg, 1:1 of the N and C-terminal half). Scale bar: 100 μm. **b–d** Quantification of dystrophin-positive fibers in the heart (**b**), gastrocnemius (GA, **c**) and diaphragm (Diaph, **d**) muscles. 10w, 10 weeks old ($n = 4$ *mdx$^{4cv}$* and 5 AAV9-iNG treated mice); 10 m, 10 months old ($n = 4$ mice /group); ****$p < 0.0001$; two-tailed unpaired *t* test. Data are mean ± s.e.m. **e** Western blot analysis of heart homogenates from 10-week-old mice with anti-dystrophin, Cas9 and Gapdh antibodies. The WT muscle lysates were loaded at 5 μg/lane while the *mdx$^{4cv}$* muscle lysates were loaded at 25 μg/lane. The samples were derived from the same experiment and the gels/blots were processed in parallel. **f** Densitometry quantification of Western blot data shown in (**e**). $n = 4$, 6 and 5 mice for WT, *mdx$^{4cv}$* and AAV9-iNG groups, respectively; one-way ANOVA with Turkey's multiple comparisons test. Data are mean ± s.e.m. **g** Western blot analysis of heart homogenates from 10-month-old mice with anti-dystrophin and Gapdh antibodies. **h** Densitometry quantification of Western blot data in (**g**). $n = 8$, 8 and 4 mice for WT, *mdx$^{4cv}$* and AAV9-iNG groups, respectively; ****$p < 0.0001$; one-way ANOVA with Turkey's multiple comparisons test. Data are mean ± s.e.m. **i** Representative sequencing trace of dystrophin transcripts of WT and *mdx$^{4cv}$* mouse hearts (10 weeks or 10 months old) with or without AAV9-iNG treatment. **j, k** Quantification of the targeted T-to-C editing efficiency in the *mdx$^{4cv}$* mouse hearts (**j**, 10 weeks; **k**, 10 months) as assayed by sequencing of dystrophin transcripts. $n = 3$ *mdx$^{4cv}$*, 5 AAV9-iNG 10 weeks and 4 AAV9-iNG 10 months; ****$p < 0.0001$; two-tailed unpaired *t* test. Data are mean ± SD.

*mdx$^{4cv}$* hearts were essentially dystrophin negative (0.03 ± 0.02%; $N = 4$; $p = 0.002$) (Fig. 5b). Dystrophin was also rescued in skeletal muscles (gastrocnemius and diaphragm) of *mdx$^{4cv}$* mice treated with AAV9-iNG, albeit the recovery was less robust as compared to that in the heart (Fig. 5c, d, and Supplementary Fig. S11). Western blot analysis showed that dystrophin was rescued in *mdx$^{4cv}$* mouse heart to 45.9 ± 11.7% of the WT level

following systemic AAV9-iNG treatment ($p = 0.01$; Fig. 5e, f). Consistent with the immunofluorescence data, Western blot showed dystrophin was restored to about 8.0 ± 2.6% of the WT level in the gastrocnemius muscle of *mdx$^{4cv}$* mice (Supplementary Fig. S12).

A group of *mdx$^{4cv}$* mice treated with intravenous administration of AAV9-iNG at 5 weeks of age were kept for 10 months to

study the long-term impact of systemic ABE editing therapy. Surprisingly, a near complete dystrophin restoration (95.9 ± 1.6%, $p < 0.0001$) was observed in the hearts of all four treated $mdx^{4cv}$ mice at 10 months of age (Fig. 5a, b and Supplementary Figs. S13–18). Dystrophin was also rescued in the skeletal muscles (gastrocnemius and diaphragm) of these older animals with a similar percentage of dystrophin-positive muscle fibers as analyzed at the 10 weeks of age (Fig. 5c, d and Supplementary Figs. S19). Western blot analysis showed near WT levels of dystrophin expression in the hearts of the 10-month-old $mdx^{4cv}$ mice treated with AAV9-iNG (Fig. 5g, h).

The heart and muscle tissues contain many different types of cells, which make it challenging to precisely determine the DNA editing efficiency in myocytes. To estimate the editing efficiency of the *Dmd* gene, we extracted the total RNA from the heart tissues treated with or without AAV9-iNG, amplified the target region by RT-PCR, and analyzed the resulting amplicons by Sanger sequencing and the BEAT program[54]. The AAV9-iNG treated $mdx^{4cv}$ hearts showed an average 32.6 ± 2.0% T-to-C editing at 10 weeks of age ($p < 0.0001$, Fig. 5i, j) and 84.6 ± 2.6% at 10 months of age ($p < 0.0001$, Fig. 5i, k).

Repeated cycles of muscle degeneration and regeneration in muscular dystrophy result in muscle fibrosis. To examine if systemic AAV9-iNG delivery could improve the histopathology of $mdx^{4cv}$ mice, we performed Trichrome staining in 10-month-old mice. As compared to WT mice, the $mdx^{4cv}$ mice showed elevated fibrosis in both diaphragm and gastrocnemius muscles and the fibrotic areas in these muscles were significantly reduced in the $mdx^{4cv}$ mice treated with AAV9-iNG (Fig. 6a–c). Consistent with previous studies that the *mdx* mice do not develop overt cardiomyopathy before 1 year old, we found that there were no significant changes in cardiac fibrosis in $mdx^{4cv}$ mice with or without AAV9-iNG treatment at 10 months of age as compared to the WT controls (Fig. 6d). AAV9-iNG treatment also significantly reduced the percentage of centrally nucleated fibers (CNF) in both diaphragm and gastrocnemius muscles at 10 weeks of age (Fig. 6e, g). By 10 months of age, the effects of AAV9-iNG treatments on CNF were blunted (Fig. 6f, h). Although we did not observe a significant difference in cross-sectional area (CSA) of muscle fibers following AAV9-iNG treatment (Fig. 6i, j, and Supplementary Fig. 20a, b), the AAV9-iNG treatment appeared to shift the fiber size distribution towards those of the WT muscles (Fig. 6k, l, and Supplementary Fig. 20c, d), particularly in gastrocnemius muscles at 10 weeks of age.

To test if systemic AAV9-iNG treatment could improve the muscle function, we measured the muscle contractility using an in vivo muscle test system[55]. Maximum plantarflexion tetanic torque was measured during supramaximal electric stimulation of the tibial nerve at 150 Hz. While the $mdx^{4cv}$ mice produced significantly reduced torque as compared to the WT controls, systemic delivery of AAV9-iNG significantly increased the tetanic torque in $mdx^{4cv}$ mice (Fig. 6m).

**The safety profile and off-target activity of AAV9-iNG treatment.** Previous studies from others and us showed that AAV-mediated delivery of CRISPR/Cas9 into neonatal mice resulted in humoral immune responses to AAV capsid but not Cas9[17,56]. In contrast, AAV-mediated delivery of CRISPR/Cas9 into adult mice evoked robust anti-Cas9 immunity[17]. Serum samples were collected to analyze the host immune responses to the AAV9 capsid and the base editor iABE-NGA. Intramuscular injection of AAV9-iNG into 5–6 weeks old $mdx^{4cv}$$mdx^{4cv}$ mice produced robust anti-AAV9 capsid (Fig. 7a) and anti-Cas9 antibodies

(Fig. 7b) at 2 weeks after injection. The anti-AAV9 titers were similar at different time points from 2 to 7 weeks post intra-muscular injection and from 7 to 9 weeks post intravenous injection (Fig. 7a). The anti-Cas9 antibody titers showed a large variation among mice at 2 weeks after intramuscular injection, but all increased to peak by 4 weeks (Fig. 7b).

We further examined the liver toxicity of AAV9-iNG treatment by measuring serum aspartate aminotransferase (AST) and alanine aminotransferase (ALT), and kidney toxicity by measuring blood urine nitrogen (BUN). As compared to WT mice, the $mdx^{4cv}$ mice showed elevated AST (Fig. 7c) and ALT (Fig. 7d). However, treatment of $mdx^{4cv}$ mice with AAV9-iNG did not further increase the serum levels of AST and ALT at either 8 weeks or 10 months of age. Measurement of BUN did not find significant changes in the treated or untreated $mdx^{4cv}$ mice (Fig. 7e).

One concern with ABE-mediated gene correction is the potential off-target activities such as gRNA mismatch tolerance, bystander editing, and off-target RNA editing. Previous studies showed that ABE can tolerate 1–2 mismatches between the gRNA and its target sites[57]. Prediction by Cas-OFFinder[58] showed that one site on chromosome 16 (Chr16_OT) has only one mismatch, two other sites have two mismatches and 55 sites have three mismatches (Fig. 7f and Supplementary Table S3). The Chr16_OT differs from the $mdx^{4cv}$ target sequence by only one C at position 12. We transfected Neuro-2a cells with ABE-NG or iABE-NGA plus the gRNA, amplified the Chr16_OT by PCR and subjected the amplicon to next generation sequencing (NGS). As shown in Fig. 7g, we did not observe significant editing of the $A^4$ in either ABE-NG or iABE-NGA transfected cells. Similarly, we analyzed the off-target site on chromosome 1 (Chr1_OT), which differs from the $mdx^{4cv}$ target sequence by an A at position 2 and a G at position 20. Again, we found that ABE-NG or iABE-NGA did not edit the $A^4$ at Chr1_OT (Fig. 7h). Finally, we analyzed four additional off-target sites that share the same 12-bp seed region for the targeting gRNA by NGS. Again, none of these sites showed above-background levels of editing activities (Supplementary Fig. S21).

Next, we analyzed the potential bystander editing at the on-target $mdx^{4cv}$ locus in the mice treated with AAV9-iNG. Since the 10-month treated mouse hearts showed a high level of dystrophin rescue, we first determined the on-target editing efficiency in these mouse hearts by NGS. As mouse hearts contain multiple different cell types, we predicted that analysis of the genomic DNA PCR products would significantly underestimate the editing efficiency. To verify this, we performed NGS of the genomic DNA PCR products from two mouse hearts receiving AAV9-iNG and exhibiting high dystrophin rescue, and detected an up to 11% edits at A4. Thus, we sequenced the RT-PCR products to estimate the editing efficiency at the on-target $mdx^{4cv}$ locus. The A at position 4 (corresponding to the T within the premature stop codon in the coding strand) was converted to G with high efficiency from all four mouse hearts (Supplementary Fig. S22). On average, 86.2 ± 2.4% A-to-G conversion was measured (Fig. 7i). At the $mdx^{4cv}$ target site, there was only one A within the editing window of 4–8, disallowing us to analyze the bystander A-to-G editing at this particular site. Another type of undesired ABE-mediated genome edits at an on-target locus is ABE-dependent cytosine-to-uracil conversion resulting in C•G to T•A mutation at that site[44,46,59]. We found that $C^6$ at the $mdx^{4cv}$ target site was edited above background with an average efficiency of 1.6 ± 0.1% (Fig. 7j).

Finally, we performed RNA-seq to characterize the transcriptome-wide RNA editing induced by AAV9-iNG in the $mdx^{4cv}$ mouse heart samples. After filtering the confident variants from control $mdx^{4cv}$ heart samples, we found 34-185 RNA editing events in the three

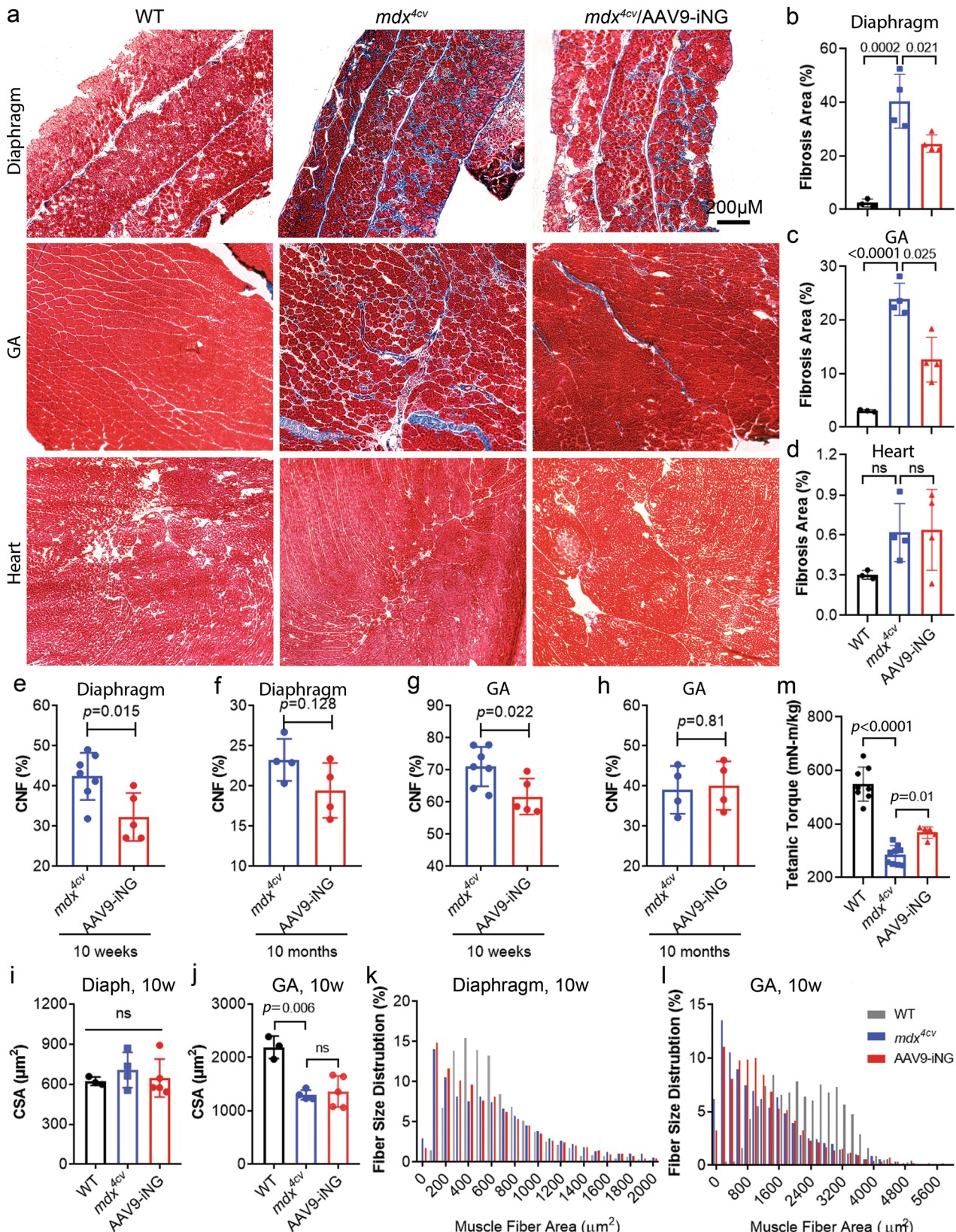

AAV9-iNG treated heart samples with 22 shared by all of them (Supplementary Fig. S23 and Supplementary Table S4). Some of the affected genes such as *Myom2*, *RyR2*, and *Myh7* play important functions in the heart, highlighting the need of further lowering the off-target RNA editing activities of ABE and temporal control of the ABE expression for in vivo disease correction.

## Discussion

Collectively, we have improved the split ABE-NG for AAV-mediated in vivo delivery by engineering a new NG PAM-interacting domain variant, a new adenine deaminase domain with higher on-target DNA editing efficiency without compromising the high fidelity of ABE-V82G, and a Gp41-1 intein split

**Fig. 6 Systemic delivery of AAV9-iNG improved histopathology and contractility in *mdx^4cv* mice. a** Trichrome staining of muscle and heart sections showing the extensive fibrosis in diaphragm and gastrocnemius (GA) muscles of *mdx^4cv* mice (10 months of age). The *mdx^4cv* mouse heart had little fibrosis at 10 months of age. Scale bar: 200 μm. **b–d** Quantification of fibrotic area of the diaphragm, gastrocnemius and heart muscles. *n* = 3 WT, 4 *mdx^4cv* and 4 AAV9-iNG treated mice. *ns* not significant; one-way ANOVA with Turkey's multiple comparisons test. Data are mean ± SD. **e–h** Measurement of CNF in the diaphragm (**e**, **f**) and gastrocnemius (**g**, **h**) muscles of *mdx^4cv* mice with or without AAV9-iNG treatment at 10 weeks (*n* = 7 untreated and 5 treated mice, (**e**, **g**) or 10 months (*n* = 4 mice/group, **f**, **h**) of age (two-tailed unpaired *t* test). Data are mean ± SD. **i–l** Muscle fiber size measurement and distribution in diaphragm and gastrocnemius muscles of the mice (*n* = 3 WT, 4 *mdx^4cv* and 5 AAV9-iNG treated) at 10 weeks of age. *ns* not significant; one-way ANOVA with Turkey's multiple comparisons test. Data are mean ± SD. **m** Tetanic torque measurements of the posterior compartment muscles of the mice (*n* = 8 WT, 9 *mdx^4cv* and 5 AAV9-iNG treated; one-way ANOVA with Turkey's multiple comparisons test). Data are mean ± SD.

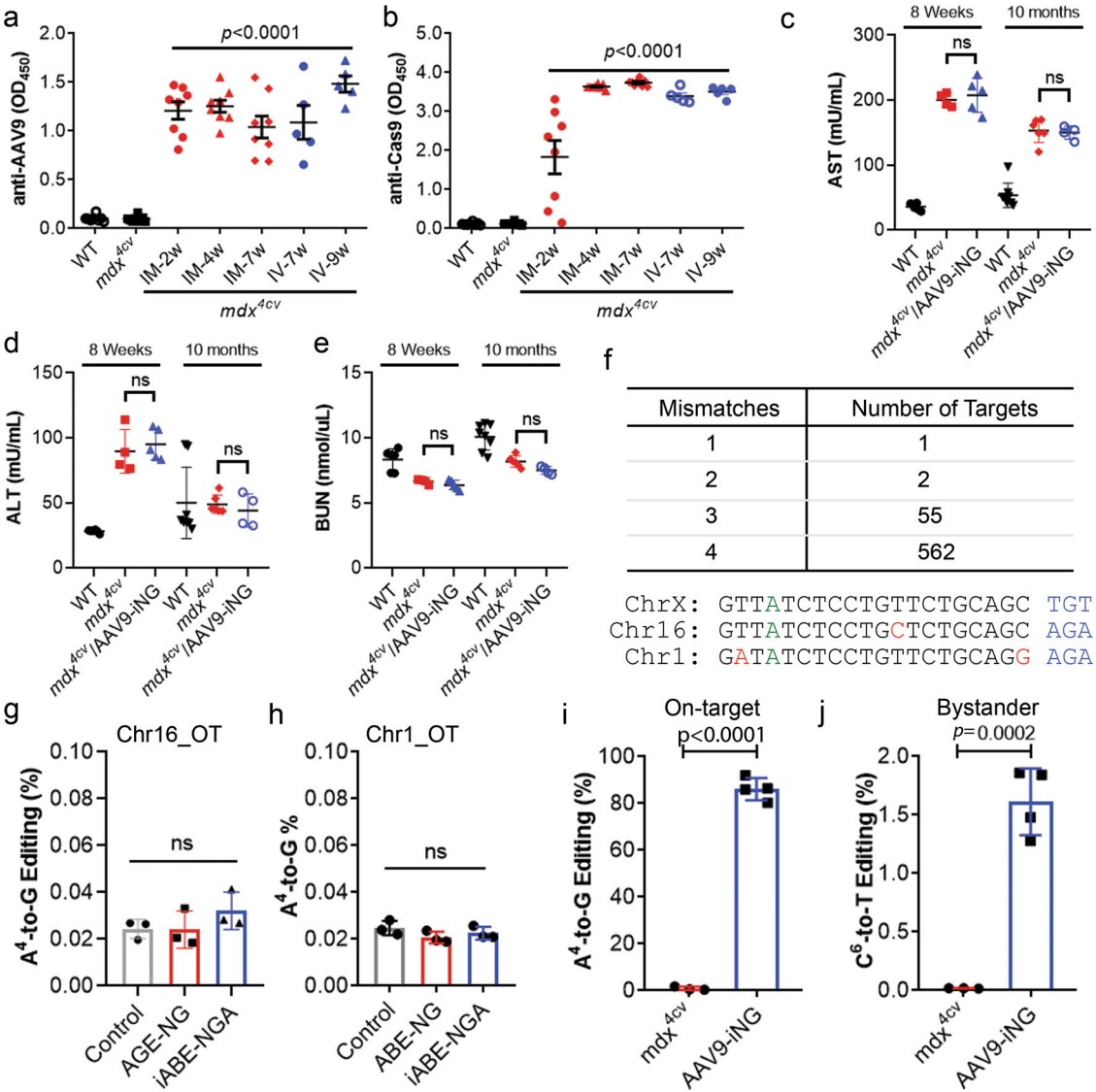

**Fig. 7 Host immune response, toxicity and off-target studies of AAV9-iNG therapy in *mdx^4cv* mice. a**, **b** Host immune response to AAV9 capsid and the base editor transgene (anti-Cas9). *IM* intramuscular injection, *IV* intravenous injection. *n* = 5 mice for IV groups and eight mice/group for the rest; one-way ANOVA with Turkey's multiple comparisons test. Data are mean ± s.e.m. **c–e** Measurements of serum AST (**c**), ALT (**d**) and BUN (**e**) of mice treated with or without AAV9-iNG. *n* = 6 WT, 4 *mdx^4cv* and 5 AAV9-iNG treated, 10 weeks; *n* = 8 WT, 6 *mdx^4cv* and 4 AAV9-iNG treated, 10 months. *ns* not statistically significant (one-way ANOVA with Turkey's multiple comparisons test). Data are mean ± SD. **f** Predicted off-target sites with different number of mismatches from the *mdx^4cv* target gRNA, and the sequences of two most similar off-target sites located on chromosome 16 and 1, respectively. **g**, **h** Quantification of deep sequencing reads of the genomic DNA PCR amplicons of the chromosome 16 off-target site (**g**) or the chromosome 1 off-target site (**h**) from Neuro-2a cells transfected with ABE-NG, iABE-NGA or control plus the gRNA. *n* = 3/group. *ns* not significant (one-way ANOVA with Turkey's multiple comparisons test). Data are mean ± SD. **i** Quantification of the A^4-to-G editing in *mdx^4cv* mice treated with or without AAV9-iNG. *n* = 3 *mdx^4cv* and 4 AAV9-iNG treated; two-tailed unpaired *t* test. Data are mean ± SD. **j** Quantification of the bystander C^6-to-T editing in *mdx^4cv* mice treated with or without AAV9-iNG. *n* = 3 *mdx^4cv* and 4 AAV9-iNG treated; two-tailed unpaired *t* test. Data are mean ± SD.

that mediates higher efficiency of protein splicing and editing. Together, these improvements allowed us to achieve widespread dystrophin rescue and functional improvement in dystrophic mice. The editing efficiency in the heart was high in $mdx^{4cv}$ mice following systemic delivery of AAV9-iNG and over 95% of cardiomyocytes were corrected to express dystrophin in $mdx^{4cv}$ hearts at 10 months of age after a single intravenous administration of AAV9-iNG at 5 weeks old. There was no obvious toxicity detected following AAV9-iNG treatment, despite the host immune response to the AAV9 capsid and ABE. This has tremendous implication for base correction of genetic cardiomyopathies.

We showed that the ecTadA* domain with the V82G mutation had significantly reduced on-target DNA editing activity as compared to the original ecTadA*. By adding the A56G mutation into the V82G variant of ecTadA*, we showed that the on-target DNA editing activity was dramatically improved without compromising the high fidelity of the V82G variant in terms of its low off-target RNA editing activity. Although the A56G_V82G variant was less efficient than the recently reported ABE8e[47], we found that ABE8e had also significantly higher bystander editing activity than the A56G_V82G variant. It is essential to consider both the on-target DNA editing efficiency and the bystander DNA editing as well as off-target RNA editing activity for in vivo applications. Ideally, the editors should have high on-target DNA editing but induce minimal bystander DNA editing and off-target RNA editing events. The A56G_V82G variant offers a reasonable balance between the editing efficiency and the editing precision.

Our study provides the proof-of-principle evidence that the iABE-NGA could effectively correct a nonsense point mutation in a mouse model of DMD. Analysis of the ClinVar database showed that 100 out of the 508 DMD point mutations could be targeted for repair by iABE-NGA (Supplementary Table S1), accounting for only a small percentage of total DMD cases. However, the applicability of iABE-NGA could be greatly expanded when considering the targeting capability of iABE-NGA to mutate the splicing sites for exon skipping in DMD[60,61]. Moreover, iABE-NGA base editing could be used for targeting other genetic diseases. An estimation from the ClinVar database showed that 16698 out of 53469 total pathogenic mutations could be targeted by iABE-NGA editing. Finally, the recent advances in engineering Cas9 variants with non-G PAM[36,37] could further increase the targeting scope.

It is of interest that the mice at 10 months after AAV9-iNG delivery appeared to show significantly higher dystrophin rescue than the mice at 10 weeks after the treatment. One plausible explanation is that the DMD cardiomyocytes with restored dystrophin expression may gain advantage for selective survival and regeneration during the development stages after delivery of AAV9-iNG. In addition, transduced cardiomyocyte-derived extracellular vesicles may deliver genetic materials such as transcripts encoding iABE-NGA into proximal un-transduced cardiomyocytes and confer base editing in those cells. Future studies will be needed to explore these possibilities.

Our study has also shown that systemic delivery of AAV9-iNG resulted in dystrophin restoration in skeletal muscles and functional improvement. As compared to cardiomyocytes, the editing efficiency in skeletal muscles was substantially lower. An obvious explanation could be that AAV9 has higher tropism towards cardiomyocytes than skeletal muscles[62]. However, other mechanisms may also be responsible for the lower editing efficiency in skeletal muscles. For example, the high rate of necrosis in skeletal muscles as compared with the relative stability of dystrophic cardiomyocytes has recently been suggested to be responsible for the rapid loss of edited genomes[63]. The dystrophic and inflammatory microenvironment in skeletal muscles may

also pose further constrains on AAV9 delivery and base editing. In addition, targeting muscle satellite cells may be required to improve the overall editing outcomes in skeletal muscle as they are constantly activated to replace injured skeletal muscle in DMD. Although AAV9 has been shown to transduce muscle satellite cells, the efficiency is relatively low[13,64–66]. Moreover, the use of a muscle-specific promoter would probably further reduce the base editing in muscle satellite cells in our present study. Future studies are required to improve the base editing efficiency in dystrophic skeletal muscles.

Recently, Liu and colleagues employed a similar intein PTS approach to split ABE and CBE at the Cys 574 split site of Cas9 and packaged into AAV for in vivo delivery, and showed ~20% editing efficiency in heart[52], lower than the editing efficiency achieved in our study. Several differences between these studies may explain the different editing efficiencies observed. First, the intein used in our study (Gp41-1) has a superfast kinetics as compared to that used by Levy et al., which allows more efficient assembly of full-length ABE (Fig. 4). Second, each half of the AAVs carries a gRNA-expressing cassette in our study, while in the study by Levy et al.[52], gRNA is present in only the C-terminal half of the Npu intein split constructs. Previous studies[16,53] and our own data (Fig. 4h) showed that the gRNA dosage affects the editing efficiency. Third, the promoters used in these studies were also different, which may drive different expression levels of ABE in heart tissues. Finally, the intrinsic difference in the gRNAs and ABE variants may have impacts on the overall editing outcomes. Nevertheless, the high editing efficiency achieved in adult dystrophic mice suggests that our optimized AAV9-iNG vectors could be used for future clinical applications.

## Methods

**Mice**. Mice (C57BL/6 J and B6Ros.Cg-$Dmd^{mdx-4Cv}$/J) were purchased from the Jackson Laboratory and housed at The Ohio State University Laboratory Animal Resources in accordance with animal use guidelines. All the experimental procedures were approved by the Institutional Animal Care and Use Committee of the Ohio State University. All mice were maintained under standard conditions of constant temperature (72 ± 4 °F), humidity (relative, 30–70%), in a specific pathogen-free facility and exposed to a 12-h light/dark cycle.

**Plasmid construction**. The pCMV-ABE7.10, pCMV-ABE-xCas9(3.7) and pCMV-ABEmax were obtained from Addgene. The NG and other mutations in the PAM domain were introduced by fusion PCR of pCMV-ABEmax and subcloned into pCMV-ABEmax. The A56G and V82G mutations were introduced into TadA* domain by fusion PCR. The CfaN minigene was synthesized by IDTdna and fused at the amino acid 573 of SpCas9-max through PCR amplification. The TadA-TadA*-SpCas9max(2-573)-CfaN fragment was PCR amplified and subcloned into pAAV under the control of meCMV promoter to generate pAAV-ABEmaxN-temp. The hU6 promoter with $mdx^{4cv}$-targeting gRNA was PCR amplified and cloned into pAAV-ABEmaxN-temp to make pAAV-ABEmaxN. The CfaC fused with SpCas9max(574-end) was generated by PCR and cloned into pAAV-ABEmaxN-temp to make pAAV-ABEmaxC. Similarly, pAAV-ABEmaxN2 and pAAV-ABEmaxC2NG with the Gp41-1 intein, and pAAV-ABEmaxN3 and pAAV-ABEmaxC3NG with the Npu intein were constructed. The $mdx^{4cv}$ gRNA and other gRNA oligos (listed in Supplementary Table S5) were annealed and ligated into pLenti-ogRNA. The $mdx^{4cv}$ reporter oligos were annealed and ligated into pLKO-puro-2A-EGFP to form the $mdx^{4cv}$ reporter plasmid as previously described[39]. All plasmids used in this study are listed in Supplementary Table S6.

**Generation of AAV particles**. AAV vectors were produced at the viral vector core of the Nationwide Children's Hospital[7]. The Gp41-1 intein split of iABE-NGA and the gRNA targeting $mdx^{4cv}$ mutation (GTTaTCTCCTGTTCTGCAGC TGT; note: the underlined PAM sequences were not included in the gRNA) or a non-targeting gRNA (GTTTaTGTCACCAGAGTAAC, the different nucleotides are highlighted in blue) expression cassettes were packaged into AAV9 capsid using the standard triple transfection protocol[67]. AAV9-iNG was titered using digital droplet PCR. Titers are expressed as DNase resistant particles per ml (DRP/ml) and rAAV titers used for injection in mice were $3.0 \times 10^{13}$ DRP/ml.

**Cell culture and transfection**. HEK293 and Neuro-2a cells were cultured in Dulbecco's modified eagle's medium (DMEM) (Corning, Manassas, VA) containing 10% fetal bovine serum (FBS) and 1% 100x penicillin-streptomycin

(10,000 U/ml, Invitrogen). Cells were plated in 6-well plates and transfected with the 2 µg plasmids (1:1 for gRNA and ABE co-transfection, or 1:1.5:1.5 for reporter, gRNA and ABE triple transfection) per well by polyethylenimine (PEI) (for HEK293) or X-tremeGENE HP DNA transfection reagent (Roche, Mannheim, Germany) (for Neuro-2a) as previously described[68].

**Flow cytometry.** At 72 h post transfection, HEK293 cells transfected with ABE were collected from 6-well plate and analyzed on Becton Dickinson LSR II with BD FACSDiva software version 8.0.1 (BD Biosciences) to determine GFP-positive cells. A total of 100,000 cell events were collected and data analysis was performed using the FlowJo 10.4 software (Tree Star, Ashland, OR, USA). The gating strategy is shown in Supplementary Fig. S24.

**Intramuscular and intravenous administration of AAV9 particles.** AAV9-iNG viral particles ($2 \times 10^{11}$ vg, 25 µl) were injected into the right gastrocnemius compartment of the male $mdx^{4cv}$ mice at 5–6 weeks of age. For systematic delivery, the male $mdx^{4cv}$ mice at 5–6 weeks of age were administered with AAV9-iNG viral particles ($1 \times 10^{14}$ vg/kg) or saline vehicle via tail vein injection.

**Serological analysis.** Blood samples were collected at various time points after intramuscular or intravenous injection. The blood samples were allowed to clot for 15 min to 30 min and centrifuged at 2300 g for 10 min at room temperature. The supernatant was collected as serum and stored at −80 °C for the biochemical assays. Measurement of ALT (BioVision Incorporated), AST (BioVision Incorporated), BUN (Arbor Assays, Michigan, USA), and cardiac Troponin I (Life Diagnostics, Inc) were performed according to the manufacturer's protocols.

**Antibody ELISA.** Antibodies against AAV9 and SpCas9 were detected by adapting previously published protocols[17,56,69]. In brief, recombinant AAV9 ($2 \times 10^9$ vg/well) and SpCas9 protein (0.27 µg/well) were diluted in 1x Coating Buffer A (BioLegend, San Diego, CA) and used to coat a 96-well Nunc MaxiSorp plate. Proteins were incubated overnight at 4 °C to adsorb to the plate. Plates were washed four times 5 min each with PBS plus 0.05% Tween-20 and then blocked with 1x Assay Diluent A (BioLegend) for 1 h at room temperature. The anti-AAV2 (A20, cat. # 03-65155, American Research Products, Inc, Waltham, MA) and anti-SpCas9 antibody (C15310258, Diagenode, Denville, NJ) was used as positive control for detection of anti-AAV9 and anti-SpCas9 antibodies, respectively. Serum samples were added in 1:50 dilution and plates were incubated for 2 h at room temperature with shaking. Plates were washed four times 5 min each and 100 µl of 1x Assay Diluent A (BioLegend) containing goat anti-mouse IgG (7076 S, 1:4000, Cell Signaling Technology, Danvers, MA) was added to each well and incubated at 1 h at room temperature. Plates were washed four times 5 min each, 100 µl of freshly mixed TMB Substrate Solution (BioLegend) was added to each well, and incubated in the dark for 20 min. The reaction was stopped by adding 100 µl 2 N $H_2SO_4$ Stop Solution. Optical density at 450 nm was measured with a plate reader.

**Muscle contractility measurements.** At 5 weeks after intravenous AAV9-iNG injection, muscle contractility was measured using an in vivo muscle test system (Aurora Scientific Inc). Mice were anesthetized with 3% (w/v) isoflurane and anesthesia was maintained by 1.5% isoflurane (w/v) during muscle contractility measurement. Maximum plantarflexion tetanic torque was measured during a train of supramaximal electric stimulations of the tibial nerve (pulse frequency 150 Hz, pulse duration 0.2 ms) using the DMA v5.501 (Aurora Scientific Inc).

**Histopathological assessment of tissues.** Mice were sacrificed at various time points, and tissues (heart, diaphragm and gastrocnemius) were harvested for histological, histochemical, biochemical, and molecular analyses. For immunohistological examinations, tissues were embedded in optimal cutting temperature (OCT, Sakura Finetek, Netherlands) compound and snap-frozen in cold isopentane for cryosectioning. The tissues were stored at −80 °C and processed for biochemical analysis and histology assessment. Frozen cryosections (7 µm) were fixed with 4% paraformaldehyde for 15 min at room temperature. After washing with PBS, the slides were blocked with 3% BSA for 1 h. The slides were incubated with primary antibodies against dystrophin (ab15277, 1:100, Abcam, Cambridge, MA) and laminin-α2 (ALX-804-190-C100, 1:100, Enzo Life Sciences Inc, Farmingdale, NY) at 4 °C for 1 h. After that, the slides were washed extensively with PBS and incubated with secondary antibodies Alexa Fluor 488 goat anti-rat IgG (A-11006, 1:400, Invitrogen, Carlsbad, CA) or Alexa Fluor 568 donkey anti-rabbit IgG (A10042, 1:400, Invitrogen, Carlsbad, CA) for 1 h at room temperature. The slides were sealed with VECTASHIELD Antifade Mounting Medium with DAPI (Vector Laboratory, Burlingame, CA). All images were taken under a Nikon Ti-E fluorescence microscope (magnification 200x) (Nikon, Melville, NY). Laminin-α2-positive and dystrophin-positive muscle fibers were counted using NIS-Elements AR version 4.50 (Nikon, Melville, NY). The amount of dystrophin positive muscle fibers is represented as a percentage of total laminin-α2-positive muscle fibers.

For trichrome staining, we used Masson's 2000 Trichrome Kit (American MasterTech, Lodi, CA). The muscle and heart sections were fixed with 4% paraformaldehyde for 1 h at room temperature. After washing with PBS, the tissue sections were stained with Masson's trichrome reagent following the manufacturer's instruction.

**Western blot analysis.** Mouse tissues from $mdx^{4cv}$ mice treated with or without AAV9-NG or AAV9-iNG were lysed with cold RIPA buffer supplemented with protease inhibitors and extracted protein samples were separated by SDS-PAGE (BioRad, 4–15%) and transferred onto Nitrocellulose membranes (0.45 µm). The rabbit polyclonal anti-dystrophin (E2660, 1:500, Spring Bioscience, Pleasanton, CA), rabbit polyclonal anti-Cas9 (C15310258-100, 1:1000, Diagenode, Denville, NJ) and rabbit monoclonal anti-Gapdh (2118 S, 14C10, 1:2000, Cell Signaling Technology, Danvers, MA) antibodies were used for immunoblotting analysis. HRP conjugated goat anti-mouse (7076 S, 1:4000) and goat anti-rabbit (7074 S, 1:4000) secondary antibodies were obtained from Cell Signaling Technology, Danvers, MA. The membranes were developed using ECL western blotting substrate (Pierce Biotechnology, Rockford, IL) and scanned by ChemiDoc XRS + system (BioRad, Hercules, CA). Western blots were quantified using Image Lab 6.0.1 software (Bio-Rad Laboratories, Hercules, CA) according to the manufacturer's instruction. The assembly efficiency of intein-split ABE was measured as the percentage of the assembled full-length ABE band intensity / (assembled full-length ABE band intensity + un-assembled fragment band intensity) on Western blot analysis.

**Extraction of genomic DNA and total RNA, PCR and Sanger sequencing.** Genomic DNA from mouse tissues and cultured HEK293 cells were extracted using DNeasy Blood & Tissue Kit (Qiagen, Germantown, MD). Total RNA was extracted from mouse tissues and HEK293 cells using Quick-RNA MiniPrep Kit (ZYMO Research, Irvine, CA). Five µg of treated RNA was used as template for first-strand cDNA synthesis by using RevertAid RT Reverse Transcription Kit (Life Technologies, Carlsbad, CA). Aliquots of the RT product were used for RT-PCR analysis of dystrophin editing. PCR reactions were carried out with 100 ng genomic DNA or cDNA in the GoTaq Master Mix (Promega) according to the manufacturer's instruction. The primers used for RT-PCR of the reporter genes and PCR of endogenous loci were listed in Supplementary Table S7. The PCR products were purified using the Wizard SV Gel and PCR Clean-up System (Promega). Purified genomic DNA and RT PCR products (100 ng) were subjected to Sanger sequencing at the Genomics Shared Resource of the Ohio State University Comprehensive Cancer Center. The sequencing data were analyzed by BEAT program[54].

**Targeted deep sequencing.** The on-target and off-target loci were first amplified by genomic DNA PCR and/or RT-PCR using gene-specific primers with Illumina adapters (primers are provided in Supplementary Table S7). The first PCR products were purified using a commercial purification kit (Promega, Madison, WI, USA), diluted, pooled, and subjected to a second round PCR with primers including the index sequences. The final PCR products were electrophoresed on an agarose gel, showing a single sharp peak. The quality and quantity were assayed using an Agilent Bioanalyzer 2100 (Genomics Shared Resource, Ohio State University Comprehensive Cancer Center). The purified amplicons were pooled and sent for sequencing using a MiSeq nano-scale flow cell (Paired-end 300 base-pair reads) at The Genomics Services Laboratory of Nationwide Children's Hospital. The FASTQ files were analyzed using CRISPResso2[70] with default parameters.

**RNA-seq experiments.** RNA library preparation was performed using NEBNext® Ultra™ II Directional (stranded) RNA Kit for Illumina (NEB #E7760L New England Biolabs) with an initial input of 100 ng ng extracted RNA per sample, measured using Qubit RNA HS reagents (#Q32852 Invitrogen) for fragmentation, cDNA synthesis and amplification. Depletion of ribosomal RNA (rRNA) was carried out with NEBNext rRNA Depletion Kit (human, mouse, rat) from New England Biolabs (#E6310X). NEBNext Multiplex oligos indexes kits (E7335L, E7500L, and E7710L) from New England Biolabs were used to barcode each library following the manufacturer protocol. RNA-seq libraries were examined using an Agilent 2100 Bioanalyzer and a High Sensitivity DNA kit (Agilent Technologies, Inc). RNA-seq libraries were sequenced on Novaseq SP Paired-End 150 bp format at The Genomics Services Laboratory of Nationwide Children's Hospital.

**RNA sequence variant calling and variant filtering.** Illumina paired-end fastq sequencing reads were processed according to GATK Best Practices for RNA-seq variant calling. In brief, reads were aligned to the mouse mm10 reference genome using STAR version 1.5.2 in two-pass mode with the parameters implemented by the ENCODE project. Picard tools (version 2.19.0) was then applied to sort and mark duplicates of the mapped BAM files. The refined BAM files were subject to split reads that spanned splice junctions, local realignment, base recalibration, and variant calling with SplitNCigarReads, IndelRealigner, BaseRecalibrator, and HaplotypeCaller tools from GATK (version 4.1.2.0), respectively. Known variants in dbSNP version 142 were used during base quality recalibration. From all called variants, downstream analyses focused solely on single-nucleotide variants (SNVs) on canonical (1–22, X, Y, and M) chromosomes. To identify variants with high confidence, we filtered clusters of at least five SNVs that were within a window of 35 bases and variants with Fisher strand values >30.0, qual by depth values <2.0 and sequencing depth <30. Base edits labeled as A-to-I comprise A-to-I edits called on the positive strand as well as T-to-C edits sourced from the negative strand, since the RNAs were converted into cDNA

before sequencing, both the nucleotide and its complementary base could be sequenced. Results obtained with our pipeline may underestimate the actual number of RNA edits occurring in cells because of the high stringency of our variant calling pipeline and potential under-representation of intronic and intergenic RNA in our experiments.

Any confident variants found in wild-type Neuro-2a cells were considered to be SNPs and were filtered out from the base-editor-transfected groups for off-target analysis. Similarly, any confident variants found in control $mdx^{4cv}$ heart samples were filtered out from the AAV9-iNG group for off-target analysis. The editing rate was calculated as the number of mutated reads divided by the sequencing depth for each site.

**Statistical analysis**. The data were expressed as mean ± the standard deviation of the mean (SD), analyzed with GraphPad Prism 8.0.1 software (San Diego, USA) and final figures were assembled with Adobe Photoshop 2020. Statistical differences were determined by two-tailed unpaired Student's $t$ test for two groups and one-way ANOVA with Turkey's post tests for multiple group comparisons using Prism 8 (Graphpad Software, La Jolla, California) with the assumption of Gaussian distribution of residuals. A $p < 0.05$ was considered to be significant.

**Reporting summary**. Further information on research design is available in the Nature Research Reporting Summary linked to this article.

## Data availability

The sequencing data that support the findings of this study have been deposited in the National Center for Biotechnology Information Sequence Read Archive (SRA) and are accessible through the BioProject accession number PRJNA673243. All other relevant data are available within the article and its Supplementary Information files or from the corresponding author upon reasonable request. Source data are provided with this paper.

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

## Acknowledgements

The authors thank the Viral Vector Core at the Nationwide Children's hospital for producing the AAV, the Analytical Cytometry Shared Resource of the Ohio State University Comprehensive Cancer Center for FACS, the Genomics Shared Resource of the Ohio State University Comprehensive Cancer Center for sequencing. R.H. is supported by US National Institutes of Health grant (R01 HL116546) and a Parent Project Muscular Dystrophy award.

## Author contributions

R.H. conceived the study, designed all plasmids, and wrote the paper. L.X. constructed plasmids and carried out the experiments in Fig. 1, Fig. 4a–d and Supplementary Figs. S2, S3. C.Z. constructed plasmids and performed the experiments in Fig. 3, Fig. 4e–h, Fig. 5a–d, Fig. 6, Fig. 7c–j and Supplementary Figs. S1, S3–S11, S13–S19a, S20–23. P.W. constructed plasmids and carried out the experiments in Fig. 2 and Fig. 3d. H.W. performed the experiments in Fig. 5e–k and Supplementary Figs. S12, S19b-c. Y.G. constructed plasmids and performed the ELSA experiments in Fig. 7a, b. Y.G. and C.Z. assisted the AAV injections, maintained the mouse colony, and coordinated mouse tissue and blood collections. W.D.A. and C.Z. performed in vivo muscle contractility experiments. JM contributed to drafting and revision of the manuscript. All authors contributed to the final version of the paper.

## Competing interests

The authors (R.H., Y.G., and L.X.) have submitted a patent application based on the results reported in this paper. All other authors confirm no other conflicts of interest.
