## [Peer Review File · Nature Communications]

Reviewers' Comments:

Reviewer #1:

Remarks to the Author:

The manuscript submitted by Xu et al. describes the optimisation of a base editor and its in vivo application to a mouse model of Duchenne muscular dystrophy. The work is of interest in terms of the engineering of the novel base editor, which adds knowledge to the field and advances the technology.

Unfortunately the manuscript is not written well. This would need addressing before acceptance for publication. As written currently, it would be difficult to reproduce the work. There are a number of shortcomings in the results as presented that would preclude acceptance for publication at this time.

The following aspects are of major concern:

1. The primary claim is that there is functional improvement in edited muscle following long-term follow-up of a single systemic injection. This is established to a limited degree in one skeletal muscle, but is not shown for cardiac muscle. Since the heart showed such high levels of dystrophin restoration, functionality should be tested. ECG and left ventricular hemodynamic assay (as per Hakim et al 2018) should be performed on treated, non-treated and control animals.
2. More in-depth analysis of the muscle (skeletal and cardiac) histology should be made. Assessment of change in levels of fibrosis, quantification of fibre diameter, quantification of number of centrally-nucleated fibres.
3. Change in weight of body-wide muscles in the long-term treatments should have been included and some assessment of animal activity/strength made.
4. The findings and conclusions made would have been strengthened by use of various doses of AAV. It is not clear whether 1×10^{14} vp/kg of each vector was injected, or whether a total of 1×10^{14} vp/kg. There are important controls missing in the treatments – each vector alone, and vectors without guides.
5. It would be important to establish the levels of dCas9 protein and RNA guide expression in skeletal and cardiac tissue.
6. It is recommended that the work is repeated with AAV8 vectors to examine whether the editing levels in skeletal muscle can be improved.
7. Since the iABEmaxNG is a new variant in the base editing tool box, rather than screening activity against the four RNA adenines previously identified as being efficiently modified by ABEmax, RNASeq should be performed to establish its off-target RNA editing activity.
8. The authors might consider writing the manuscript from the angle of optimising the base editor and construct design and then providing proof-of-concept of activity in mdx4cv mouse.
9. To add weight to the findings, an in vivo comparison to the base editors developed by Liu et al 2019, and those described by Levy et al 2020 should be made.
10. The use of an inactivating split-tein mutation would provide an important control.

Minor concerns would include:

1. Dystrophin gene should be written as DMD or Dmd gene throughout the manuscript.
2. The gating for the FACS analysis is not shown in supplementary data.
3. No description of assumptions or corrections made are given for the statistical analyses.
4. The titration method used for AAV preps is different for AAV9-NG, and for AAV9-iNG.
5. The western blot for dystrophin in Fig 2 is of poor quality.
6. Xu et al 2019 is cited twice as ref 50 and 67.

Reviewer #2:

Remarks to the Author:

In this manuscript, "Efficient precise in vivo base editing in adult dystrophic mice", Xu et al. delivered the adenine base editors (ABE) which can recognize a broader NG PAM, using adeno-

associated viral vectors, to a mouse model of Duchenne muscular dystrophy (DMD) to correct the pathogenic mutation and recover the muscle functional deficiencies. This manuscript demonstrate the broader therapeutic potential of base editing in adult animals. However, additional compares, studies, and analyses are required to validate their claims.

Major Concerns:

1. The off-target analyses in this manuscript are very hasty. For applying the CRISPR tools to gene therapy, one of the major concern is off-targets. To my surprise, the authors only inspected two potential off-target sites in vitro, did not even perform inspecting in vivo, then they claimed ABE-NG and iABE-NG show low tolerance to mismatches. Whole-genome sequencing should be performed in different organs of treated mice to carefully assess the effects of off-targets, since the authors used tail vein injection to globally express the editor. Also, the RNA off-targets of ABEs are not well examined. They only tested five RNA sites in vitro which were previously shown to be highly modified by ABEmax. Previous study (Grunewald et al., 2019) has observed that miniABEmax(V82G) still have approximately fourfold and threefold higher numbers of edited adenines relative to background, though it performed well in previously identified as being highly edited RNA adenines. Thus, RNA-seq should also be performed to examine the RNA off-targets.

2. The description about re-engineering ABE-NG is confusing. First, previous study (Grunewald et al., 2019) have characterized the RNA off-targets of miniABEmax(V82G) and miniABEmax(A56G), and found miniABEmax(A56G) didn't perform well. Why the authors chose the combination of this two mutation? The authors should also prove the combination of this two mutation has the advantages over the single mutation V82G. Second, previous study (Niu et al., 2020) has shown that engineered SpCas9 based on mutations from both xCas9 and Cas9-NG has better recognition of NGH PAM, the authors did not cite or discuss this previous study. Further, the rationale behind the choice of amino acid mutations chosen for this study needs to be further reinforced.

3. The authors claimed that the intein with fast rate of PTS they used can improve the efficiency of split ABE, which is one of the reasons that they have better efficiency than other group. However, no experimental data can support this conclusion. Univariate experimentation should be perform to compare the gp41-1 intein with Npu intein.

Minor Concerns:

1. Why the result between the Fig. 1c, d and Fig. 1h, i is so different? Actually, they are similar experiments.

2. Line 162 – 164. Why RNA editing activity of ABE may affect the base editing outcomes?

3. Line 215 – 220. Please show the data to support your conclusion.

4. Line 358 – 361. High off-target RNA editing activity can't be the reason for low editing efficiency. In fact, the higher editing efficiency, the more RNA editing off-targets can be detected in previous study (Grunewald et al., 2019).

Reference

Grunewald, J., Zhou, R.H., Iyer, S., Lareau, C.A., Garcia, S.P., Aryee, M.J., and Joung, J.K. (2019). CRISPR DNA base editors with reduced RNA off-target and self-editing activities. *Nature Biotechnology* 37, 1041-1048.

Niu, Q.F., Wu, S.Q., Li, Y.S., Yang, X.X., Liu, P., Xu, Y.P., and Lang, Z.B. (2020). Expanding the scope of CRISPR/Cas9-mediated genome editing in plants using an xCas9 and Cas9-NG hybrid. *J Integr Plant Biol* 62, 398-402.

We thank the reviewers for their constructive and insightful comments. Below is a point-by-point response to each individual comments.

Reviewer #1 (Remarks to the Author):

The manuscript submitted by Xu et al. describes the optimisation of a base editor and its in vivo application to a mouse model of Duchenne muscular dystrophy. The work is of interest in terms of the engineering of the novel base editor, which adds knowledge to the field and advances the technology. Unfortunately the manuscript is not written well. This would need addressing before acceptance for publication. As written currently, it would be difficult to reproduce the work. There are a number of shortcomings in the results as presented that would preclude acceptance for publication at this time.

Response: We appreciate the Reviewer #1 for the acknowledgement of our work being of interest. We have performed additional experiments and revised the manuscript as suggested. In brief, we have rationally improved the split ABE-NG for AAV-mediated in vivo delivery by engineering a new NG PAM-interacting domain variant, a new adenine deaminase domain with higher on-target DNA editing efficiency without compromising the high fidelity of ABE-V82G, and a Gp41-1 intein split that mediates higher efficiency of protein splicing and editing. Together, these improvements allowed us to achieve close to complete dystrophin rescue in dystrophic mouse heart.

The following aspects are of major concern:

1. The primary claim is that there is functional improvement in edited muscle following long-term follow-up of a single systemic injection. This is established to a limited degree in one skeletal muscle, but is not shown for cardiac muscle. Since the heart showed such high levels of dystrophin restoration, functionality should be tested. ECG and left ventricular hemodynamic assay (as per Hakim et al 2018) should be performed on treated, non-treated and control animals.

Response: We have performed additional histopathological analysis of the muscle following systemic AAV9-iNG delivery. As shown in **Figure 6**, the muscle fibrosis in both diaphragm and gastrocnemius muscles were significantly decreased following AAV9-iNG treatment. However, similar to the *mdx* mice, the *mdx*^{4cv} mice do NOT present overt cardiomyopathy before one year old. Consistently, we showed that the cardiac fibrosis in 10-month-old *mdx*^{4cv} mice was not significantly increased as compared to age-matched controls (**Figure 6**). The lack of cardiomyopathy in 10-month-old *mdx*^{4cv} mice does not allow us to test the functional improvement in the heart following treatment. In our ongoing studies, we are breeding *mdx*^{4cv} mice with utrophin^{-/-} mice to produce *mdx*^{4cv}/utrophin^{-/-} mice and then test if AAV9-iNG could improve the dystrophic cardiomyopathy in these new animal model. We will be glad to report the results in our follow-up studies.

2. More in-depth analysis of the muscle (skeletal and cardiac) histology should be made. Assessment of change in levels of fibrosis, quantification of fibre diameter, quantification of number of centrally-nucleated fibres.

Response: We have quantified fibrosis, fiber diameter and percentage of centrally-nucleated fibers. The new data is provided in **Figure 6**.

3. Change in weight of body-wide muscles in the long-term treatments should have been included and some assessment of animal activity/strength made.

Response: Thank you for the comment. We have quantified the muscle fiber size and measured the *in vivo* plantarflexion muscle contractility (this measurement is to assess the function of a group of muscles including gastrocnemius, soleus, plantaris etc.). The data is presented in **Figure 6**.

4. The findings and conclusions made would have been strengthened by use of various doses of AAV. It is not clear whether 1×10^{14} vp/kg of each vector was injected, or whether a total of 1×10^{14} vp/kg. There are important controls missing in the treatments – each vector alone, and vectors without guides.

Response: In our preliminary experiment, we tested two different doses (5×10^{13} and 1×10^{14} vg/kg) and the higher dose provided higher dystrophin restoration (Supplementary Figure S3). This dosage range is in line with the literature about systemic delivery of AAV9. The dosage we reported is the total AAV9, 1 to 1 ratio of the N and C-terminal halves. We also showed that the control experiment using AAV9-iNG with a non-target gRNA failed to rescue dystrophin expression (Supplementary Figure S3).

5. It would be important to establish the levels of dCas9 protein and RNA guide expression in skeletal and cardiac tissue.

Response: We performed Western blot and showed the expression of iABE-NGA (as indicated by anti-Cas9 antibody) in both skeletal muscle and heart tissues (Figure 5e and Supplementary Figure S12).

6. It is recommended that the work is repeated with AAV8 vectors to examine whether the editing levels in skeletal muscle can be improved.

Response: Thank you for the suggestion. We will test different AAV serotypes including AAV8 in our future studies to see if the editing levels could be improved in skeletal muscles.

7. Since the iABEmaxNG is a new variant in the base editing tool box, rather than screening activity against the four RNA adenines previously identified as being efficiently modified by ABEmax, RNASeq should be performed to establish its off-target RNA editing activity.

Response: Thank you for the suggestion. We have performed RNAseq to compare the off-target RNA editing activity following the method described in *Nature* 571, 275–278 (2019). The results showed that the transcriptome-wide off-target RNA editing was greatly diminished to the background level (**Fig. 3C**).

8. The authors might consider writing the manuscript from the angle of optimising the base editor and construct design and then providing proof-of-concept of activity in mdx4cv mouse.

Response: Thank you. We have revised the manuscript as suggested.

9. To add weight to the findings, an *in vivo* comparison to the base editors developed by Liu et al 2019, and those described by Levy et al 2020 should be made.

Response: We appreciate the comments. We performed additional *in vitro* experiments and demonstrated that our intein split outperformed the one described by Levy et al. 2020 (the data is shown in **Figure 4**). We also compared the ecTadA* domain with those reported by Richter et al., (*Nat Biotechnol* 38, 883–891 2019) and Gaudelli et al. (*Nat Biotechnol*, 38, 892–900, 2020) and the data showed that all these newly reported ABE variants conferred high level of on-target DNA editing with the ABE8e displaying the highest activity (**Supplementary Figure S1**). However, the ABE8e also induced higher level of off-target RNA editing activity within the editing window (**Supplementary Figure S1**) as reported in Richter et al., (*Nat Biotechnol* 38, 883–891 2019). Our rationally designed ecTadA* reached a reasonable degree of balance in the activity and fidelity, which is important for *in vivo* applications.

10. The use of an inactivating split-tein mutation would provide an important control.

Response: Thank you for the comments. We have tested this by swapping the Gp41-1 intein split with the Npu intein split (e.g. by combining the Gp41-1 N-terminal half with the Npu C-terminal half, or vice versa), and the data showed that the intein-mediated protein splicing is essential for the highly efficient editing in these split constructs (**Figure 4h**).

Minor concerns would include:

1. Dystrophin gene should be written as DMD or Dmd gene throughout the manuscript.

Response: Corrected.

2. The gating for the FACS analysis is not shown in supplementary data.

Response: Added.

3. No description of assumptions or corrections made are given for the statistical analyses.

Response: We have added the following statement to the Method section: The data were expressed as mean \pm the standard deviation of the mean (SD) and analyzed with GraphPad Prism 8.0.1 software (San Diego, USA). Statistical differences were determined by two-tailed unpaired Student's t test for two groups and one-way ANOVA

with Turkey's post tests for multiple group comparisons using Prism 8 (Graphpad Software, La Jolla, California) with the assumption of Gaussian distribution of residuals. A p value less than 0.05 was considered to be significant. Specific methods of statistics are presented in the figure legends. Scatter plots are used in all graphs.

4. The titration method used for AAV preps is different for AAV9-NG, and for AAV9-iNG.

Response: That's true. The AAV vectors were made by the Nationwide Children's Hospital Viral Vector Core and they recently adopted a new method for titrating AAV preparations.

5. The western blot for dystrophin in Fig 2 is of poor quality.

Response: The Western blot have been repeated and the new data are shown.

6. Xu et al 2019 is cited twice as ref 50 and 67.

Response: Corrected.

Reviewer #2 (Remarks to the Author):

In this manuscript, "Efficient precise in vivo base editing in adult dystrophic mice", Xu et al. delivered the adenine base editors (ABE) which can recognize a broader NG PAM, using adeno-associated viral vectors, to a mouse model of Duchenne muscular dystrophy (DMD) to correct the pathogenic mutation and recover the muscle functional deficiencies. This manuscript demonstrate the broader therapeutic potential of base editing in adult animals. However, additional compares, studies, and analyses are required to validate their claims.

Response: We appreciate the Reviewer #2 for the statement that our work demonstrates the broader therapeutic potential of base editing in adult animals. We have performed additional studies to substantiate our conclusions, in particular related to the functional characterization and off-target analysis.

Major Concerns:

1. The off-target analyses in this manuscript are very hasty. For applying the CRISPR tools to gene therapy, one of the major concern is off-targets. To my surprise, the authors only inspected two potential off-target sites in vitro, did not even perform

inspecting in vivo, then they claimed ABE-NG and iABE-NG show low tolerance to mismatches. Whole-genome sequencing should be performed in different organs of treated mice to carefully assess the effects of off-targets, since the authors used tail vein injection to globally express the editor. Also, the RNA off-targets of ABEs are not well examined. They only tested five RNA sites in vitro which were previously shown to be highly modified by ABEmax. Previous study (Grunewald et al., 2019) has observed that miniABEmax(V82G) still have approximately fourfold and threefold higher numbers of edited adenines relative to background, though it performed well in previously identified as being highly edited RNA adenines. Thus, RNA-seq should also be performed to examine the RNA off-targets.

Response: We agree that the off-target is one of the major concerns for CRISPR-based genome editing therapeutics. The remarkable fidelity of ABE in mouse embryos has been firmly established by whole-genome sequencing (Please see Lee et al., *Communication Biology*, 3, 19; Lee, et al., *Nature Communications*, 9: 4804; Liu et al., *Nature communications*, 9: 2338), in which ABE and gRNA were microinjected into one-cell mouse embryos. While in our study, we delivered the AAV9-ABE/gRNA into young adult *mdx^{4cv}* mice, and whole-genome sequencing is not expected to produce reliable variant calling in highly heterogeneous adult animal heart samples.

However, previous studies indeed demonstrated that ABEmax induces transcriptome-wide off-target RNA editing activity. Therefore, we performed RNA-seq to evaluate the transcriptome-wide off-target RNA editing activity of iABE-NG. The results are shown in **Figure 3** and **Supplementary Figure S21**. Using the method described in *Nature* 571, 275–278 (2019), we showed that the transcriptome-wide off-target RNA editing was greatly diminished to the background level (**Fig. 3C**).

2. The description about re-engineering ABE-NG is confusing. First, previous study (Grunewald et al., 2019) have characterized the RNA off-targets of miniABEmax(V82G) and miniABEmax(A56G), and found miniABEmax(A56G) didn't perform well. Why the authors chose the combination of this two mutation? The authors should also prove the combination of this two mutation has the advantages over the single mutation V82G. Second, previous study (Niu et al., 2020) has shown that engineered SpCas9 based on mutations from both xCas9 and Cas9-NG has better recognition of NGH PAM, the authors did not cite or discuss this previous study. Further, the rationale behind the choice of amino acid mutations chosen for this study needs to be further reinforced.

Response: Our studies with miniABEmax(V82G)-NG to edit *mdx^{4cv}* mutation showed that miniABEmax(V82G)-NG has significantly decreased on-target editing efficiency as compared to miniABEmax-NG (**Figure 3b**). We thus aimed to improve the on-target editing efficiency of miniABEmax(V82G)-NG with the goal to also maintain its high fidelity. From the crystal structure of *S. aureus* TadA-tRNA, we noticed that A56G is located near the enzymatic pocket around the substrate tRNA and in Grunewald's paper, this mutation does not significantly affect the off-target RNA editing activity but appears to increase the on-target editing activity. We thus generated the double mutant miniABEmax(A56G_V82G) and found that that introduction of A56G mutation in the miniABEmax(V82G)-NG dramatically increased the on-target editing activity (see

Figure 3b), while RNA-seq showed that this double mutant maintains low off-target RNA editing activity (**Figure 3c and d**). We have revised the text accordingly to better describe the rationale.

Thanks for pointing out the Niu et al.'s paper. We have added this reference.

3. The authors claimed that the intein with fast rate of PTS they used can improve the efficiency of split ABE, which is one of the reasons that they have better efficiency than other group. However, no experimental data can support this conclusion. Univariate experimentation should be perform to compare the gp41-1 intein with Npu intein.

Response: Thank you very much for this great comment. We have performed additional experiments and our data demonstrated that the fast rate of PTS used in our study clearly improved the assembly of full-length iABE-NGA and increased the editing efficiency. Furthermore, we showed that the two gRNAs (each in the N and C-terminal halves) are better than the split with only one gRNA in the N-terminal half but not the C-terminal half as reported in other group. These new data are provided in **Figure 4e-h**.

Minor Concerns:

1. Why the result between the Fig. 1c, d and Fig. 1h, i is so different? Actually, they are similar experiments.

Response: It was due to different x-axis setting. We have updated the Figures.

2. Line 162 – 164. Why RNA editing activity of ABE may affect the base editing outcomes?

Response: Dysregulated RNA editing has been shown to lead to grave outcomes such as autoimmune diseases (eg. Cell Rep 23: 50-57, 2018) and tumors (Cell Rep 13: 267-276). It is conceivable that the elevated transcriptome-wide RNA editing could lead to production of autoantigens and thus triggers immune elimination of the edited cells in vivo. Although beyond the scope of our current study, the in vivo consequences of the transcriptome-wide RNA off-target editing activity induced by base editors warrant careful future studies. To reduce confusion, we have removed these sentences.

3. Line 215 – 220. Please show the data to support your conclusion.

Response: The manuscript has been updated.

4. Line 358 – 361. High off-target RNA editing activity can't be the reason for low editing efficiency. In fact, the higher editing efficiency, the more RNA editing off-targets can be detected in previous study (Grunewald et al., 2019).

Response: Please see response to point 2.

Reviewers' Comments:

Reviewer #1:

Remarks to the Author:

The authors have answered the reviewers concerns well. The manuscript reads in a much clearer way because of the drastic changes made.

There are still a number of minor concerns that need addressing before acceptance for publication. Page numbers refer to tracked changes PDF.

1. Line 8 of abstract change 'the' to 'a'
2. State what the change in skeletal muscle dystrophin expression is in abstract.
3. Reference is needed for last sentence of page 3.
4. The introduction needs to include a description of the number of cases of DMD caused by point mutations and in particular those caused by a C to a T change.
5. Throughout the results terms such as 'small', 'slightly better', 'substantially reduced', 'substantially higher' are used. Add numbers and where significant changes, add the p value.
6. Page 8. It is confusing why in some circumstances you use whole genome sequencing and in others you look at only 4 previously described off target sites. Also it is not clear why you do this analysis in HEK cells rather than mouse cells.
7. Page 8 - remove 'during review of our manuscript'
8. Page 8 and also page 10 - clarity is needed on how you tested mdx-4cv editing - was this in cells (and which cells) using delivered plasmid, or in vivo?
9. Page 9 The claim at the end of 1st paragraph is too strong. Additional screening against many other sequences would be required to validate such a claim.
10. Page 10 It is not clear how you calculate the efficiency of assembly.
11. Only one p value is given throughout the results section. P values need to be given throughout.
12. Page 21 - what number of mice were analysed using RT-PCR and sequencing?
13. Page 22, last line of results. Expand on those 32 shared RNA edits.
14. Page 25. References are needed for previous studies showing gRNA dose and editing efficiency
15. The patient applicability of base editing using this construct will be extremely low. This needs to be mentioned.
16. More is needed on the plasmid used in the reporter assay. The cells used and transfection reagents are not currently described.
17. Methodologies using Neuro2As are not included at all.
18. Fig S2 B) what is the PCR analysis of and what does the arrow indicate?
19. Figs S4, S5 and S14 - what does 'mouse number shown in yellow' refer to?
20. Fig S12 - these look like different membranes.
21. Fig S19 - why wasn't the diaphragm analysed in western blot?
22. Fig 2 - stats bars as given are confusing for the statistical analyses used.
23. Fig. 2 - change A6 to A6 with 6 superscript
24. On editorial policy checklist, Neuro2A cells are not listed in eukaryotic cell lines.

Reviewer #2:

Remarks to the Author:

The manuscript has been improved subsequently with this revision, but I insist further work is required to make it more convincing.

1. Though the fidelity of ABE has been demonstrated in mouse embryos, it's a different issue in treatment because ABE is expressed continuously by AAV. What's more, the sgRNA-depend off-targets vary from site to site. If the authors hesitate to perform whole-genome sequencing, they should at least inspect more potential off-target sites in vitro and in vivo. I would like suggest using the CasOT (Xiao et al., 2014) to check potential off-target sites that have mismatches only

in the non-seed region of sgRNA.

2. The significance between miniABE(V82G)-NG and miniABE(GG)-NG in Fig. 3c is trustless. The authors should also show the expression level of the editors in RNA-seq to prove the big deviation between replicates is not caused by experimental variations (such as the dose of plasmid delivery).

We thank the reviewers for their constructive and insightful comments. Below is a point-by-point response to each individual comments.

Reviewer #1 (Remarks to the Author):

The authors have answered the reviewers concerns well. The manuscript reads in a much clearer way because of the drastic changes made.

There are still a number of minor concerns that need addressing before acceptance for publication. Page numbers refer to tracked changes PDF.

Response: We appreciate the Reviewer #1 for the statement that our revised manuscript reads in a much clearer way and has answered the reviewers concerns well. We have further revised our manuscript to fully address the minor concerns raised.

1. Line 8 of abstract change 'the' to 'a'

Response: Corrected.

2. State what the change in skeletal muscle dystrophin expression is in abstract.

Response: Added “with up to 15% rescue in skeletal muscle fibers.”

3. Reference is needed for last sentence of page 3.

Response: Added the reference.

4. The introduction needs to include a description of the number of cases of DMD caused by point mutations and in particular those caused by a C to a T change.

Response: Added “In particular, 174 out of 508 pathogenic point mutations for DMD are due to G:C to A:T conversion (Supplementary Table S1), which could potentially be targeted by ABE editing”.

5. Throughout the results terms such as 'small', 'slightly better', 'substantially reduced', 'substantially higher' are used. Add numbers and where significant changes, add the p value.

Response: Thank you. We have added the numbers and p values in the text or on the figures.

6. Page 8. It is confusing why in some circumstances you use whole genome sequencing and in others you look at only 4 previously described off target sites. Also it is not clear why you do this analysis in HEK cells rather than mouse cells.

Response: We characterized the RNA off-target activities (which are caused by gRNA independent interaction of the TadA domain with cellular RNA) in mouse cells using

RNA-seq, and then we attempted to verify the low RNA off-target activities of miniABE-GG. Since previous studies characterizing the RNA off-target activities of ABEs were primarily conducted in human cells, we chose several well-known RNA off-target sites in human cells for comparison, similar to the studies by Joung and his colleague (Grunewald et al., 2019a; Grunewald et al., 2019b).

7. Page 8 - remove 'during review of our manuscript'

Response: revised as “Two groups recently reported a new generation of ABEs through directed evolution”

8. Page 8 and also page 10 - clarity is needed on how you tested mdx-4cv editing - was this in cells (and which cells) using delivered plasmid, or in vivo?

Response: Page 8, it was revised to “To directly compare miniABE(GG) with ABE8.17, ABE8.20 and ABE8e, we fused each of them with SpCas9-NG and tested their activities for editing the *mdx*^{4cv} target site using the reporter assay in Neuro-2a cells”.

Page 10, it was revised to “we quantified the T-to-C conversion of the *mdx*^{4cv} stop codon in Neuro-2a cells using the reporter assay”.

9. Page 9 The claim at the end of 1st paragraph is too strong. Additional screening against many other sequences would be required to validate such a claim.

Response: We have revised the sentence as “Thus, we chose miniABE(GG) for the *in vivo* studies due to its high efficiency and relatively high precision.”.

10. Page 10 It is not clear how you calculate the efficiency of assembly.

Response: The assembly efficiency of intein-split ABE was measured as the percentage of the assembled full-length ABE band intensity / (assembled full-length ABE band intensity + un-assembled fragment band intensity) on Western blot analysis. This has been added into the Methods under “Western blot analysis” section.

11. Only one p value is given throughout the results section. P values need to be given throughout.

Response: The p values have now been added for all studies in the text and/or on the figures.

12. Page 21 - what number of mice were analysed using RT-PCR and sequencing?

Response: Each data point shown in Figure 7i corresponds to an individual mouse (3 control and 4 AAV9-iNG treated). The mouse IDs for this analysis are: control group: 1983, 1986 and 1987; AAV9-iNG group: 1976, 1982, 1984 and 1985.

13. Page 22, last line of results. Expand on those 32 shared RNA edits.

Response: After increasing the filter value of sequencing depth for SNV calling (see response to Reviewer 2, point 2), the shared RNA edits were 22. The 22 shared RNA edits are now listed in the new **Supplementary Table S3**.

14. Page 25. References are needed for previous studies showing gRNA dose and editing efficiency

Response: Added the references (Hakim et al., 2018; Min et al., 2019).

15. The patient applicability of base editing using this construct will be extremely low. This needs to be mentioned.

Response: The iABE-NGA could be used to correct 100 out of 508 DMD point mutations, accounting for only a small percentage of total DMD cases. However, the applicability of iABE-NGA could be greatly expanded if we consider the targeting capability of iABE-NGA to mutate the splicing sites for exon skipping in DMD cases (Gapinske et al., 2018; Winter et al., 2019). Moreover, iABE-NGA base editing could be used for targeting other genetic diseases. An estimation from the ClinVar database showed that 16698 out of 53469 total pathogenic mutations could be targeted by iABE-NGA editing. This has been added into the Discussion section.

16. More is needed on the plasmid used in the reporter assay. The cells used and transfection reagents are not currently described.

Response: The plasmid used in the reporter assay was based on our previous publication (see ref(Wang et al., 2020)), and the description of the plasmid, cells used and transfection reagents is added in the method.

17. Methodologies using Neuro2As are not included at all.

Response: Sorry for missing the information. We have now added the Neuro-2a related work in methods.

18. Fig S2 B) what is the PCR analysis of and what does the arrow indicate?

Response: The PCR was designed to detect the point mutation induced by ABE editing. The arrow indicates the correct PCR product with the desired point mutation. We have revised Fig. S2 and updated the legend.

19. Figs S4, S5 and S14 - what does 'mouse number shown in yellow' refer to?

Response: Sorry for the confusion. The figure legends for Figs. S4, S5 and S14 have been updated.

20. Fig S12 - these look like different membranes.

Response: Correct. These are different membranes because we did not want to perform stripping and re-probing with a different antibody on the same blot. But all these membranes were prepared in parallel to minimize sample loading variation.

21. Fig S19 - why wasn't the diaphragm analysed in western blot?

Response: We embedded the diaphragm muscle into O.C.T. for immunofluorescence and histology. Due to its limited availability, we did not attempt to perform Western blot using the O.C.T. embedded tissue.

22. Fig 2 - stats bars as given are confusing for the statistical analyses used.

Response: The statistical bars were updated in Fig. 2. Each new base editor variant was compared to the ABE-NG group by using One-way ANOVA with Turkey's post tests.

23. Fig. 2 - change A6 to A6 with 6 superscript

Response: Thank you for pointing this out. We have now updated Fig. 2.

24. On editorial policy checklist, Neuro2A cells are not listed in eukaryotic cell lines.

Response: We have added Neuro-2a cells into the listed on the editorial policy checklist.

Reviewer #2 (Remarks to the Author):

The manuscript has been improved subsequently with this revision, but I insist further work is required to make it more convincing.

Response: We appreciate the Reviewer #2 for the statement that our revised manuscript has been improved. We have performed additional experiments and analysis to address the concerns as below.

1. Though the fidelity of ABE has been demonstrated in mouse embryos, it's a different issue in treatment because ABE is expressed continuously by AAV. What's more, the sgRNA-depend off-targets vary from site to site. If the authors hesitate to perform whole-genome sequencing, they should at least inspect more potential off-target sites in vitro and in vivo. I would like suggest using the CasOT (Xiao et al., 2014) to check potential off-target sites that have mismatches only in the non-seed region of sgRNA.

Response: Thank you for suggesting the use of CasOT to perform off-target analysis and identify potential off-target sites that have mismatches only in the non-seed region of sgRNA. Unfortunately, we could not access the online link to the CasOT program (<http://eendb.zfgenetics.org/casot/>).

Thus, we utilized CRISPOR (<http://crispor.tefor.net/>), which was widely used by many investigators in the genome editing field. Based on the CRISPOR analysis, we identified a total of 11 potential off-target sites with mismatches only in the non-seed region of gRNA (Supplementary Data 1). Among these, six sites do not have “A” located within the targeting window (position 4-8). Thus, we performed targeted deep sequencing in Neuro-2a cells transfected with iABE-NGA/gRNA and mouse heart tissues with or without AAV injection for the rest 5 potential off-target sites. As shown in **Supplementary Figure S21**, we observed no above-background editing activities in either in cells with transient transfection or AAV-long term transduction of iABE-NGA (note: one off-target site on chromosome 5 was omitted because it did not have enough reads for analysis).

These new data further support the high fidelity of the iABE-NGA base editor. On behalf of all authors, we greatly appreciate this reviewer’s suggestion for conducting the additional off-target analysis.

2. The significance between miniABE(V82G)-NG and miniABE(GG)-NG in Fig. 3c is trustless. The authors should also show the expression level of the editors in RNA-seq to prove the big deviation between replicates is not caused by experimental variations (such as the dose of plasmid delivery).

Response: We appreciate the reviewer’s concern on the variations in the SNV calling analysis of RNAseq reads (**Fig. 3c**). We must point out that it is common to observe very large variations among replicate samples with SNV calling from RNAseq experiments (for example, please see the Figure 1c in Nat Biotechnol. 2019 (Grunewald et al., 2019b). Early studies published in Nature by (Zhou et al., 2019) only used n=2-3 for similar analysis (Figs. 1, 2 and 4).

We acknowledge the fact that the original samples were all sent for the RNA-seq analyses and thus we could not conduct the retrospective RT-PCR or Western blot assays to test if there are any potential variations in the transfection efficiency. However, we performed re-analysis of the RNAseq data. By increasing filter value of sequencing depth from 10 (in our original analysis) to 30 (see **the figure below**), which would filter out SNVs with lower confidence, we showed that the variation in the samples became smaller and the statistical significance increased. The Figure 3c is updated.

Further supporting the fidelity of the miniABE(GG)-NG is shown in Fig. 3d., where we performed detailed testing of several known off-target RNA editing sites in human cells.

Collectively, these data demonstrated that the miniABE(GG)-NG has significantly reduced off-target RNA editing activities as compared to miniABE-NG.

References

- Gapinske, M., Luu, A., Winter, J., Woods, W.S., Kostan, K.A., Shiva, N., Song, J.S., and Perez-Pinera, P. (2018). CRISPR-SKIP: programmable gene splicing with single base editors. *Genome biology* 19, 107.
- Grunewald, J., Zhou, R., Garcia, S.P., Iyer, S., Lareau, C.A., Aryee, M.J., and Joung, J.K. (2019a). Transcriptome-wide off-target RNA editing induced by CRISPR-guided DNA base editors. *Nature* 569, 433-437.
- Grunewald, J., Zhou, R., Iyer, S., Lareau, C.A., Garcia, S.P., Aryee, M.J., and Joung, J.K. (2019b). CRISPR DNA base editors with reduced RNA off-target and self-editing activities. *Nature biotechnology* 37, 1041-1048.
- Hakim, C.H., Wasala, N.B., Nelson, C.E., Wasala, L.P., Yue, Y., Louderman, J.A., Lessa, T.B., Dai, A., Zhang, K., Jenkins, G.J., *et al.* (2018). AAV CRISPR editing rescues cardiac and muscle function for 18 months in dystrophic mice. *JCI insight* 3.
- Min, Y.L., Li, H., Rodriguez-Caycedo, C., Mireault, A.A., Huang, J., Shelton, J.M., McAnally, J.R., Amosii, L., Mammen, P.P.A., Bassel-Duby, R., *et al.* (2019). CRISPR-Cas9 corrects Duchenne muscular dystrophy exon 44 deletion mutations in mice and human cells. *Science advances* 5, eaav4324.
- Wang, P., Xu, L., Gao, Y., and Han, R. (2020). BEON: A Functional Fluorescence Reporter for Quantification and Enrichment of Adenine Base-Editing Activity. *Molecular therapy : the journal of the American Society of Gene Therapy*.
- Winter, J., Luu, A., Gapinske, M., Manandhar, S., Shirguppe, S., Woods, W.S., Song, J.S., and Perez-Pinera, P. (2019). Targeted exon skipping with AAV-mediated split adenine base editors. *Cell Discov* 5, 41.

Zhou, C., Sun, Y., Yan, R., Liu, Y., Zuo, E., Gu, C., Han, L., Wei, Y., Hu, X., Zeng, R., *et al.* (2019). Off-target RNA mutation induced by DNA base editing and its elimination by mutagenesis. *Nature* 571, 275-278.

Reviewers' Comments:

Reviewer #1:

Remarks to the Author:

The authors have addressed the reviewers comments very well and the manuscript is improved as a result. There are just a couple of very minor concerns:

In line 137, change 'plant' to 'plants'.

In line 288/9 - please state which muscle in particular are being referred to here.

Reference should be made to Bengtsson et al (2020) Mol Ther 29(3) in the discussion around higher editing in the heart in comparison to skeletal muscle.

Reviewer #2:

Remarks to the Author:

The authors have addressed all of my concerns.

We are grateful to the reviewers for their timely review of our manuscript and glad to hear that both reviewers are happy with our responses to their previous critiques. Below is a point-by-point response to each individual comments.

Reviewer #1 (Remarks to the Author):

The authors have addressed the reviewers comments very well and the manuscript is improved as a result.

Response: We appreciate the Reviewer #1 for the positive statement on our revised manuscript.

There are just a couple of very minor concerns:
In line 137, change 'plant' to 'plants'.

Response: Corrected.

In line 288/9 - please state which muscle in particular are being referred to here.

Response: The skeletal muscles were specified.

Reference should be made to Bengtsson et al (2020) Mol Ther 29(3) in the discussion around higher editing in the heart in comparison to skeletal muscle.

Response: We have added the reference by Bengtsson et al (2020) Mol Ther 29(3) in the discussion around higher editing in the heart in comparison to skeletal muscle.

Reviewer #2 (Remarks to the Author):

The authors have addressed all of my concerns.

Response: Thank you very much! We are happy to hear that all of Reviewer #2's concerns have been addressed.

References